# Two-photon imaging in mice shows striosomes and matrix have overlapping but differential reinforcement-related responses

Bernard Bloem[1,2†], Rafiq Huda[2,3†], Mriganka Sur[2,3], Ann M Graybiel[1,2*]

[1]McGovern Institute for Brain Research, Massachusetts Institute of Technology, Cambridge, United States; [2]Department of Brain and Cognitive Sciences, Massachusetts Institute of Technology, Cambridge, United States; [3]Picower Institute for Learning and Memory, Massachusetts Institute of Technology, Cambridge, United States

**Abstract** Striosomes were discovered several decades ago as neurochemically identified zones in the striatum, yet technical hurdles have hampered the study of the functions of these striatal compartments. Here we used 2-photon calcium imaging in neuronal birthdate-labeled Mash1-CreER;Ai14 mice to image simultaneously the activity of striosomal and matrix neurons as mice performed an auditory conditioning task. With this method, we identified circumscribed zones of tdTomato-labeled neuropil that correspond to striosomes as verified immunohistochemically. Neurons in both striosomes and matrix responded to reward-predicting cues and were active during or after consummatory licking. However, we found quantitative differences in response strength: striosomal neurons fired more to reward-predicting cues and encoded more information about expected outcome as mice learned the task, whereas matrix neurons were more strongly modulated by recent reward history. These findings open the possibility of harnessing in vivo imaging to determine the contributions of striosomes and matrix to striatal circuit function.
DOI: https://doi.org/10.7554/eLife.32353.001

**\*For correspondence:**
graybiel@MIT.EDU

†These authors contributed equally to this work

**Competing interests:** The authors declare that no competing interests exist.

## Introduction

The striatum, despite its relatively homogeneous appearance in simple cell stains, is made up of a mosaic of macroscopic zones, the striosomes and matrix, which differ in their input and output connections and are thought to allow specialized processing by physically modular groupings of striatal neurons (*Crittenden et al., 2016*; *Fujiyama et al., 2011*; *Gerfen, 1984*; *Graybiel and Ragsdale, 1978*; *Jiménez-Castellanos and Graybiel, 1989*; *Langer and Graybiel, 1989*; *Lopez-Huerta et al., 2016*; *Salinas et al., 2016*; *Smith et al., 2016*; *Stephenson-Jones et al., 2016*; *Walker et al., 1993*; *Watabe-Uchida et al., 2012*). Particularly striking among these modules are the striosomes (also called patches), which are distinct from the surrounding matrix and its constituent modules by differential expression of neurotransmitters, receptors and many other gene expression patterns, including those related to dopaminergic and cholinergic transmission (*Banghart et al., 2015*; *Brimblecombe and Cragg, 2015*; *Brimblecombe and Cragg, 2017*; *Crittenden and Graybiel, 2011*; *Cui et al., 2014*; *Flaherty and Graybiel, 1994*; *Gerfen, 1992*; *Graybiel, 2010*; *Graybiel and Ragsdale, 1978*). Striosomes in the anterior striatum have strong inputs from particular regions related to the limbic system, including parts of the orbitofrontal and medial prefrontal cortex (*Eblen and Graybiel, 1995*; *Friedman et al., 2015*; *Gerfen, 1984*; *Ragsdale and Graybiel, 1990*) and, at subcortical levels, the bed nucleus of the stria terminalis (*Smith et al., 2016*) and basolateral

amygdala (*Ragsdale and Graybiel, 1988*). The striosomes are equally specialized in their outputs: they project directly to subsets of dopamine-containing neurons of the substantia nigra (*Crittenden et al., 2016*; *Fujiyama et al., 2011*) and, via the pallidum, to the lateral habenula (*Rajakumar et al., 1993*; *Stephenson-Jones et al., 2016*). By contrast, the matrix and its constituent matrisomes receive abundant input from sensorimotor and associative parts of the neocortex (*Flaherty and Graybiel, 1994*; *Gerfen, 1984*; *Parthasarathy et al., 1992*; *Ragsdale and Graybiel, 1990*), and project via the main direct and indirect pathways to the pallidum and non-dopaminergic pars reticulata of the substantia nigra (*Flaherty and Graybiel, 1994*; *Giménez-Amaya and Graybiel, 1991*; *Kreitzer and Malenka, 2008*), universally thought to modulate movement control (*Albin et al., 1989*; *Alexander and Crutcher, 1990*; *DeLong, 1990*).

This contrast in connectivity between striosomes and the surrounding matrix highlights the possibility that striosomes, which physically form three-dimensional labyrinths within the much larger matrix, could serve as limbic outposts within the large sensorimotor matrix. The question of what the actual functions of striosomes are, however, remains unsolved. Answering this question has importance for clinical work as well as for basic science: striosomes have been found, in post-mortem studies, to be selectively vulnerable in disorders with neurologic and neuropsychiatric features (*Crittenden and Graybiel, 2016*; *Saka et al., 2004*; *Sato et al., 2008*; *Tippett et al., 2007*). Ideas about the functions of striosomes have ranged from striosomes serving as the critic in actor-critic architecture models (*Doya, 1999*), to their generating responsibility signals in hierarchical learning models (*Amemori et al., 2011*), to their being critical to motivationally demanding approach-avoidance decision-making prior to action (*Friedman et al., 2017*, *2015*), and to other functions (*Brown et al., 1999*; *Crittenden et al., 2016*). However, the technical difficulties involved in reliably identifying and recording the activity of striosomal neurons have been exceedingly challenging; striosomes are too small to yet be detected by fMRI, and their neurons have remained unrecognizable in in vivo electrophysiological studies with the exception of those identifying putative striosomes by combinations of antidromic and orthodromic stimulation (*Friedman et al., 2017*, *2015*). With the development of endoscopic calcium imaging (*Bocarsly et al., 2015*; *Carvalho Poyraz et al., 2016*; *Luo et al., 2011*) and 2-photon imaging of deep-lying structures (*Dombeck et al., 2010*; *Howe and Dombeck, 2016*; *Kaifosh et al., 2013*; *Lovett-Barron et al., 2014*; *Mizrahi et al., 2004*; *Sato et al., 2016*), combined with the use of genetic mouse models that allow direct visual identification of selectively labeled neurons, identifying functions of these specialized striatal zones should be within reach.

Here we report that we have developed a 2-photon microscopy protocol for simultaneously examining the activity of striosomal and matrix neurons in the dorsal caudoputamen of behaving head-fixed mice in which we used fate-mapping to label preferentially striosomal neurons by virtue of their early neurogenesis relative to that of matrix neurons (*Fishell and van der Kooy, 1987*; *Graybiel, 1984*; *Graybiel and Hickey, 1982*; *Hagimoto et al., 2017*; *Kelly et al., 2017*; *Newman et al., 2015*; *Taniguchi et al., 2011*). Key to this work was achieving dense, permanent labeling of not only striosomal cell bodies, but also their striosome-bounded neuropil. We accomplished this differential labeling by pulse-labeling with tamoxifen during the generation time of the spiny projection neurons (SPNs) of striosomes using Mash1(Ascl1)-CreER;Ai14 driver lines with induction at embryonic day (E) 11.5 (*Kelly et al., 2017*). This method allowed striosomal detection based on the labeling of SPN cell bodies as well as the rich neuropil labeling of the striosomes, capitalizing on the fact that SPN processes of striosome and matrix compartments rarely cross striosomal borders (*Bolam et al., 1988*; *Lopez-Huerta et al., 2016*; *Walker et al., 1993*). Thus even though only a fraction of striosomal neurons were tagged, it was possible, because of the restricted neuropil labeling generated by their local processes, to identify neurons as being inside striosomes and, concomitantly, to identify clearly neurons as lying outside of the zones of neuropil labeling, in the matrix.

With this method, we compared the activity patterns of striosomal and matrix neurons related to multiple elementary aspects of striatal encoding as mice performed a classical conditioning task. By having cues signaling different reward delivery probabilities, we tested whether striosomes and matrix differentially encode changes in expected outcome and received rewards (*Amemori et al., 2015*; *Bayer and Glimcher, 2005*; *Bromberg-Martin and Hikosaka, 2011*; *Friedman et al., 2015*; *Keiflin and Janak, 2015*; *Matsumoto and Hikosaka, 2007*; *Oyama et al., 2010*, *2015*; *Schultz, 2016*; *Schultz et al., 1997*; *Stalnaker et al., 2012*; *Watabe-Uchida et al., 2017*, *2012*). By imaging day by day during the acquisition and overtraining periods of the task, we asked

whether these patterns changed in systematic ways with experience. Finally, we tested the effect of reward history on the activity patterns of current trials, given reports that strong reward-history activity has been found in sites considered to be directly or indirectly connected with striosomes (*Bromberg-Martin et al., 2010*; *Hamid et al., 2016*; *Tai et al., 2012*).

We demonstrate that neurons visually identified as being within striosomes or within the extra-striosomal matrix have considerable overlap in their response properties during all phases of task performance. Thus, striosomes and matrix share common features related to simple reward processing and manifest acquisition of responses to different task events as a result of reward-based learning. The activities of neurons in the striosome and matrix compartments differed, however, in their relative emphases on different task epochs. Striosomal neurons more strongly encoded reward prediction, and matrix neurons more strongly encoded reward history. These findings suggest that neurons in striosomes and matrix can be differentially tuned by reinforcement contingencies both during learning and during subsequent performance. This work opens the opportunity for future functional understanding of striosome-matrix architecture by in vivo microscopy combined with selective tagging of neurons with known developmental origins, an opportunity that will be valuable conceptually in linking developmental programs to circuit function, and in the study of both normal animals and those representing models of disease states.

## Results

To detect striosomes, we performed experiments in Mash1-CreER;Ai14 mice, following the method of *Kelly et al., 2017*. This method takes advantage of the finding that Mash1 is a differential driver of the striosomal lineage during the ~E10-E13 window of neurogenesis of striosomes in mouse (*Kelly et al., 2017*). We injected pregnant Mash1-CreER;Ai14 dams with tamoxifen at E11.5, in the middle of this neurogenic phase of striosomal development. This treatment led to the permanent expression in the resulting offspring of tdTomato in cells being born at the time of induction. We found strong tdTomato labeling of striosomes in the striatal regions of the caudoputamen that we examined (*Figure 1*, *Figure 1—figure supplement 1*). Critically, this labeling marked not only the cell bodies of the striosomal neurons, but also their local processes, which were confined to the neuropil as confirmed histologically in initial immunohistochemical experiments (*Figure 1*). These experiments demonstrated that the clusters of labeled neurons and their neuropil corresponded to striosomes, as evidenced by the close match between the zones of tdTomato neuropil labeling and mu-opioid receptor 1 (MOR1)-rich immunostaining (*Table 1*) (*Kelly et al., 2017*; *Tajima and Fukuda, 2013*). We also observed sparsely distributed tdTomato-labeled neurons outside of MOR1-labeled striosomes, scattered in the extra-striosomal matrix, but they never exhibited patchy neuropil labeling.

For in-vivo experiments, we used 2-photon microscopy to image the striatum of 5 striosome-labeled mice that had received unilateral intrastriatal injections of AAV5-hSyn-GCaMP6s and had been implanted with cannula windows and a headplate (*Figure 2A*). Each mouse was trained on a classical conditioning task in which two auditory tones (1.5 s duration each) were associated with reward delivery by different probabilities (tone 1, 80% vs tone 2, 20%) (*Figure 2B*). Inter-trial intervals were 7 ± 1.75 s. With training, mice began to lick in anticipation of the reward, and the amount of this anticipatory licking became greater when cued by the tone indicating a high probability (80%) of reward (*Figure 2C*). We calculated a learning criterion based on the anticipatory lick rates during the two cues and the subsequent delay period (0.5 s). Mice exhibiting a divergence in anticipatory licking for the two cues for at least two out of three consecutive sessions were considered as trained (*Figure 2D*). We performed imaging during training (n = 3; task acquisition) and after this criterion had been reached (n = 5; criterion). Two mice were trained for an additional five sessions (overtraining), in which we imaged the same fields of view as in the criterion phase.

### Imaging of striosomes

Clusters of tdTomato-positive neurons were clearly visible in vivo in the 2-photon microscope at 40x magnification, and the neuropil of these neurons delimited zones in which many dendritic processes could be identified (*Figure 3*). We simultaneously recorded transients in striosomal and matrix neurons from fields of view with clear striosomes. In all animals, we could see at least two different striosomes, from which we imaged at least five different non-overlapping fields of view. In some

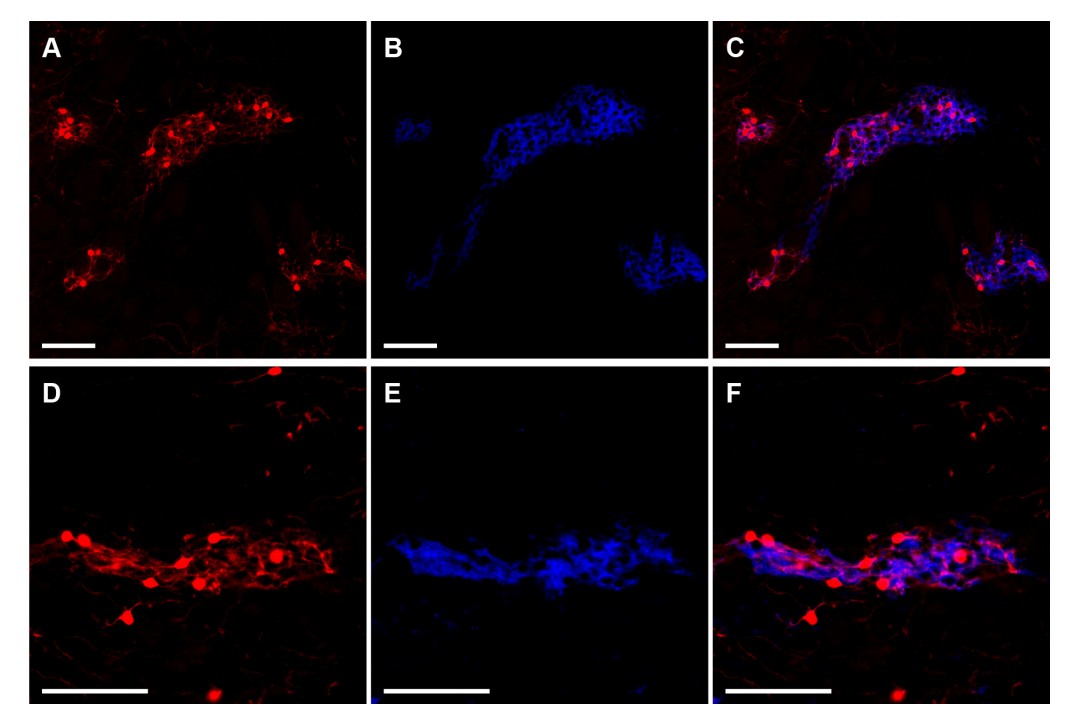

**Figure 1.** Striosomes are labeled with tdTomato in Mash1-CreER;Ai14 mice that received tamoxifen at E11.5. Images illustrate two examples (rows) of striosomal labeling of cell bodies and neuropil by tdTomato (**A,D,** red) as verified by MOR1 immunostaining identifying striosomes (**B,E,** blue). Merged images show overlap of tdTomato and MOR1 labeling (**C,F**). Scale bars indicate 100 µm.

DOI: https://doi.org/10.7554/eLife.32353.002

The following figure supplement is available for figure 1:

**Figure supplement 1.** Striosome labeling in Mash1-CreER;Ai14 mice injected with tamoxifen at E11.5.

DOI: https://doi.org/10.7554/eLife.32353.003

instances, we could see two different striosomes in one field of view. In the entire data set, we imaged 1867 neurons in striosomes and 4453 in the matrix. Because striosomes form parts of extended branched labyrinths, it was possible to follow some striosomes through ±100 µm in depth, and across ±800 µm in the field of view. During training, we rotated through the fields of view, but after the training criterion had been reached, we recorded activity in unique non-overlapping fields of view (2704 neurons, of which 727 were in striosomes; between 252 and 782 neurons per mouse; *Table 2*).

To control for small but significant differences in GCaMP6s expression (*Table 3*) between striosomes and matrix, we calculated ΔF/F as: ΔF/F = $F_t$ – $F_0$ / $F_0$ ($F_t$: fluorescence at time t; $F_0$: baseline fluorescence). We quantified the mean, standard deviation and maximum values of the ΔF/F signal during the baseline periods to test for potential differences in the signal-to-noise ratio of our recordings, but did not observe differences between striosomal and matrix neurons (*Table 3*).

**Table 1.** Overlap of striosomes outlined using tdTomato and MOR1.

| | | MOR1 | |
| --- | --- | --- | --- |
| | | **Positive** | **Negative** |
| **tdTomato** | Positive | 14.2%±1.3% | 2.0%±0.3% |
| | Negative | 3.7%±0.6% | 80.2%±1.9% |

MOR1 test-retest error rate = 2.4%
tdTomato test-retest error rate = 2.3%

DOI: https://doi.org/10.7554/eLife.32353.004

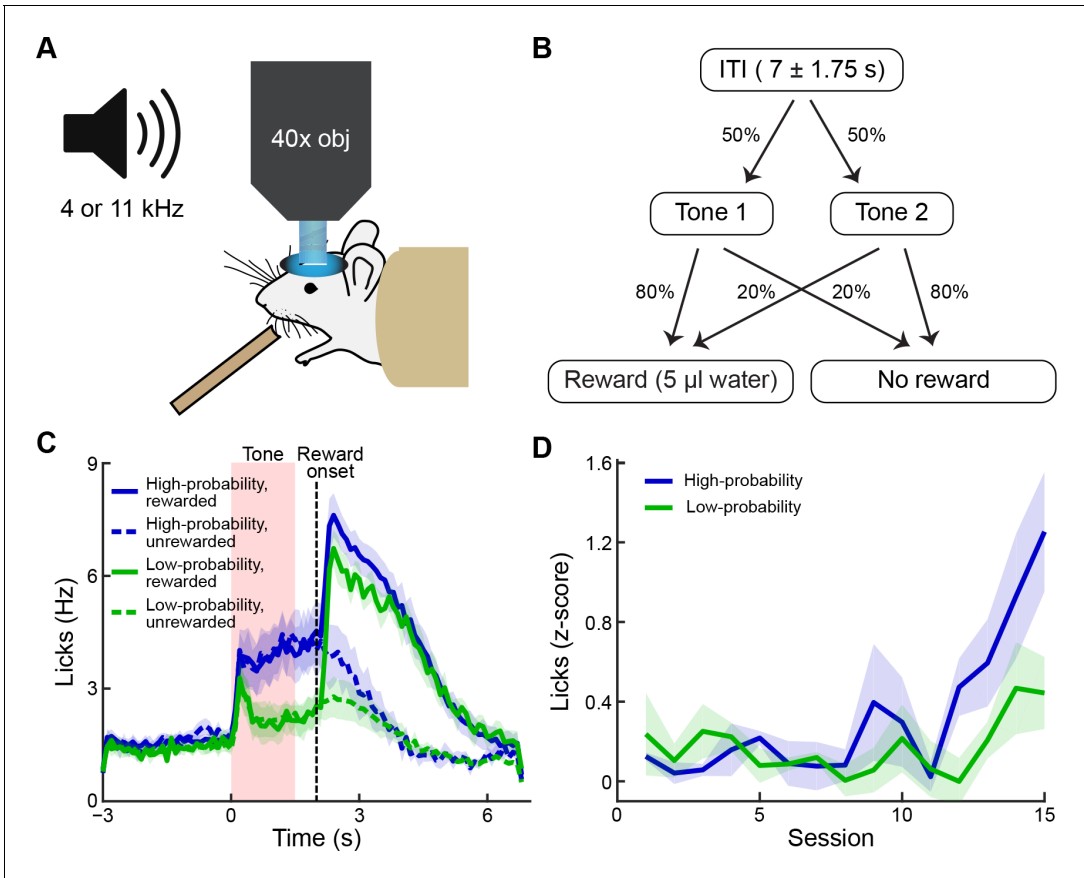

**Figure 2.** Behavioral task and performance. (**A**) The striatum was imaged during conditioning sessions in which tones predicted reward delivery. (**B**) Two tones (4 and 11 kHz) were played (1.5 s duration) and were associated with distinct reward probabilities (80% or 20%). After a 0.5 s delay, reward could be delivered. Inter-trial interval durations varied from 5.25 to 8.75 s. (**C**) Frequency of licking after training, averaged over five mice (±SEM). Anticipatory licking was significantly higher during the presentation of the high-probability tone (blue) than during the presentation of the low-probability tone (green). After reward delivery, licking rates were elevated for several seconds (solid lines: rewarded trials; dotted lines: unrewarded trials). (**D**) Licking during the tone and reward delay, shown as z-scores calculated relative to the 2 s baseline period preceding the tone, during training sessions (average of 3 mice). Mice began to exhibit differences in levels of anticipatory licking between the two cues after 11–12 sessions. Animals were considered to be trained when they exhibited significantly higher anticipatory licking during the high-probability tone (blue) than during the low-probability tone (green) in 2 out of 3 consecutive sessions. Shading represents SEM.
DOI: https://doi.org/10.7554/eLife.32353.005

## Striatal neurons exhibit heightened activity during different task epochs

As an initial approach to our data, we analyzed the overall fluorescence for every session in trained animals by averaging the frame-wide fluorescence (*Figure 4A*). Both cues evoked large responses in the neuropil signal, which were calculated as z-scores based on the mean signal and its standard deviation during a 1 s period before cue onset. These signals were larger for the high-probability cue. After reward delivery, there was a prolonged, strong activation that peaked at around 3 s after reward delivery (*Figure 4B*). To determine more precisely the nature of this activation, we aligned neuronal responses in the rewarded trials to the tone onset, to the first lick after reward delivery and to the end of the licking bout (*Figure 4B*). This analysis demonstrated that, in addition to the tone response, there was an additional increase in the signal during the post-reward licking period, and that this signal increased over time, peaked at the time of the last lick, and then subsided.

Next, we analyzed single-cell activity to investigate the neural dynamics of task encoding by the striatal neurons. In particular, we asked whether the prolonged activation seen in the frame-wide fluorescence signal was also visible in single neurons, or whether individual neurons were active during specific task events. Neuronal firing as indicated by the calcium transients was sparse during the

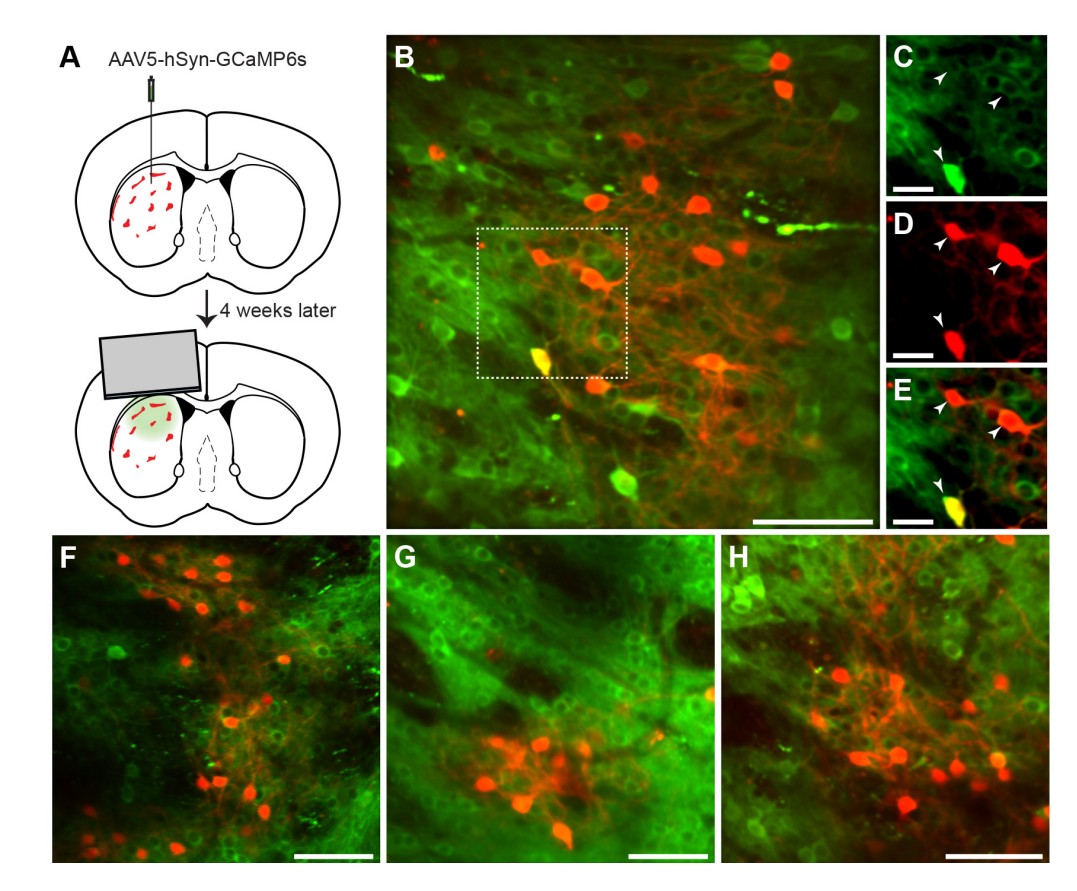

**Figure 3.** In vivo 2-photon calcium imaging of identified striosomes and matrix. (**A**) Mash1-CreER;Ai14 mice were injected with AAV5-hSyn-GCaMP6s and 4 weeks later were implanted with a cannula. (**B**) Image of a striosome acquired with the 2-photon microscope, illustrating tdTomato labeling in red and GCaMP in green (scale bar: 100 μm) in the striatum of a trained mouse. (**C–E**) Higher magnification images of the region indicated in B (scale bar: 10 μm), shown for individual green (C), red (D) and merged (E) channels. Arrowheads indicate double-labeled cells. (**F–H**) Representative examples of striosomes imaged in three other trained mice (scale bars: 100 μm).

DOI: https://doi.org/10.7554/eLife.32353.006

task, but we found that individual neurons were active for particular events during the task (*Figure 4D,G*). For instance, the red color-coded neuron illustrated in *Figure 4C and D* became active soon after tone onset, whereas the neuron color-coded in gray fired during the post-reward licking period. The timing of their activities with respect to specific trial events seemed relatively stable, resembling what has been reported before for neurons in the striatum of behaving rodents by recording and analyzing spike activity (*Bakhurin et al., 2017*; *Barnes et al., 2011*; *Gage et al.,*

**Table 2.** Numbers of recorded neurons per mouse.

| | Mouse | | | | | |
| --- | --- | --- | --- | --- | --- | --- |
| | 1 | 2 | 3 | 4 | 5 | Total |
| Number of neurons | 587 | 782 | 252 | 426 | 657 | 2704 |
| Striosomal neurons | 218 (37.1 %) | 214 (27.4 %) | 41 (16.3 %) | 77 (18.1 %) | 177 (26.9 %) | 727 (26.9 %) |
| Matrix neurons | 369 (62.9 %) | 568 (72.6 %) | 211 (83.7 %) | 349 (81.9 %) | 480 (73.1 %) | 1977 (73.1 %) |
| tdTomato-positive neurons in striosomes | 33 (5.6 %) | 21 (2.7 %) | 11 (4.4 %) | 13 (3.1 %) | 33 (5.0 %) | 111 (4.1 %) |
| tdTomato-negative neurons in striosomes | 182 (31.0 %) | 191 (24.4 %) | 30 (11.9 %) | 60 (14.1 %) | 134 (20.4 %) | 597 (22.1 %) |
| tdTomato-positive neurons outside of striosomes | 3 (0.5 %) | 2 (0.3 %) | 0 (0.0 %) | 4 (0.9 %) | 10 (1.5 %) | 19 (0.7 %) |

DOI: https://doi.org/10.7554/eLife.32353.007

**Table 3.** Baseline fluorescence and ΔF/F values for striosomal and matrix neurons.

| | Cell type | | | |
| --- | --- | --- | --- | --- |
| | **Striosomal** | **In striosomal neuropil** | **tdTomato labeled** | **Matrix** |
| Baseline fluorescence | 290.0 (8.5) *** | 274.9 (8.1) *** | 337.2 (27.9) | 364.5 (6.8) |
| ΔF/F baseline mean | 11.3 (0.7) | 11.9 (0.7) | 9.3 (1.9) | 11.9 (0.4) |
| ΔF/F baseline standard deviation | 37.2 (1.3) | 38.2 (1.4) | 33.5 (3.5) | 38.5 (0.7) |
| ΔF/F baseline maximum | 250.6 (9.9) | 259.3 (11.2) | 216.2 (22.8) | 255.2 (6.0) |

***$p < 0.001$.

DOI: https://doi.org/10.7554/eLife.32353.008

*2010*; *Jog et al., 1999*; *Rueda-Orozco and Robbe, 2015*). To determine task encoding by single neurons at a population level, we defined task-modulated neurons as those that were significantly active, according to Wilcoxon sign-rank tests during the cue, reward licking and post-licking epochs of the task (see Materials and methods). Altogether, 38.2% of the striatal neurons imaged in our samples were task-modulated. Of these, most (85%) were active during only one of the three task epochs. Among task-modulated neurons, most were selectively active during the post-reward licking period (57%), but substantial numbers of neurons were also active during the tone presentation (17%) or after the licking had stopped (11%, *Figure 4E,F*).

For population analyses, we calculated z-scores for the neuronal responses using the mean and the standard deviation of the 1 s baseline period preceding tone onsets. Analysis of session-averaged population responses of neurons selectively active during these three epochs demonstrated a similar sequence of neuronal events as the sequence that we found with analysis of the frame-wide fluorescence signals. The activation of a small group of neurons after cue onset was followed by a prolonged increase in the responses of neurons active during the post-reward licking period (*Figure 4F,G*). This population activity ramped up until mice stopped licking, then quickly subsided (*Figure 4F*). The analysis of single-cell responses also identified a group of neurons that became maximally active just after the end of licking. Grouping neurons based on the epoch during which they were active and sorting responses within each group by the timing of their peak session-averaged activity exposed a tiling of task time by neurons active in each of the three epochs (*Figure 4G*).

To determine the temporal specificity of responses during the post-reward licking period for individual neurons, we compared them to the same responses shuffled for each neuron by substituting responses in a given trial with the response in the same trial from a randomly selected task-modulated neuron recorded simultaneously during the same session (*Figure 4—figure supplement 1A*). To quantify the trial-to-trial variability in responses, we computed a reliability index as the mean correlation of responses in all pairwise combinations of trials (*Rikhye and Sur, 2015*). Shuffling the data decreased response reliability, without affecting the mean peak responses, and increased the standard deviation of peak times (*Figure 4—figure supplement 1B–D*). In addition, we measured the ridge-to-background ratio, which quantifies the mean response magnitude surrounding response peaks relative to other time points (*Harvey et al., 2012*). We found that the ratio was higher for observed data as compared to the shuffled data (*Figure 4—figure supplement 1E*). Together, these analyses indicate temporal specificity in the responses of individual neurons and suggest that the prolonged ramping of population activity observed during the post-reward licking period was produced by individual neurons being active within different specific time intervals during licking, and not by them being active throughout the licking period.

## Encoding of reward-predicting tones is stronger in striosomes than in the matrix

To dissociate the specific contributions of striosomes and matrix to task encoding, we again first compared aggregate GCaMP6s neuronal responses in both striatal compartments. We drew regions of interest (ROIs) around striosomes defined by tdTomato neuropil labeling and around nearby regions of the matrix in the same field of view with similar overall intensity of fluorescence and size, and compared the total amount of fluorescence from these regions. Both striosomes and

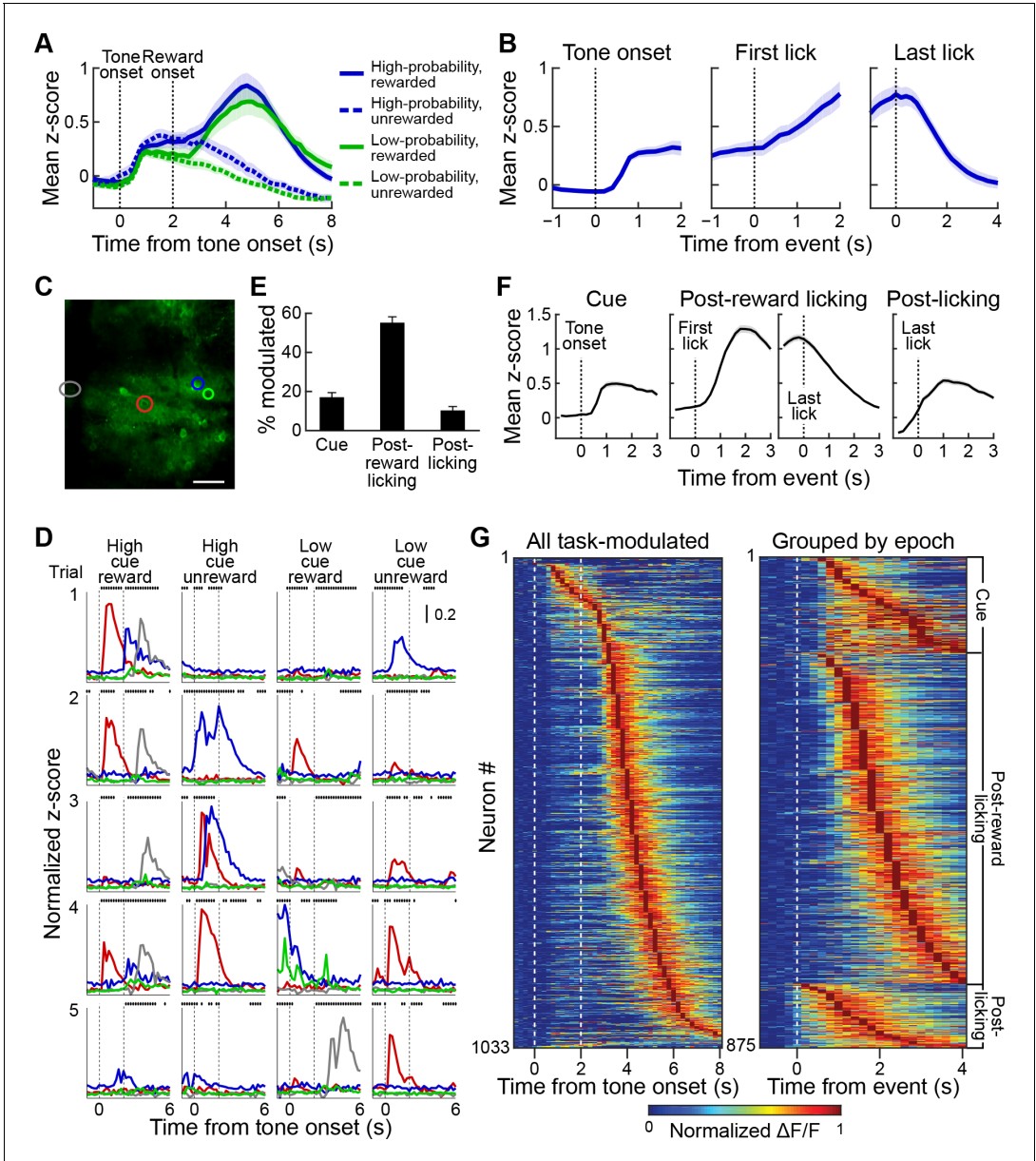

**Figure 4.** Striatal activity during reward-predicting cues and during post-reward period. (**A**) Aggregate neuropil calcium signal in all four trial types (blue: high-probability cue; green: low-probability cue; solid line: rewarded trials; dotted line: unrewarded trials). Shading represents SEM. (**B**) Neuropil activation aligned to tone onset (left), first lick after reward delivery (middle) and last lick (right). Only rewarded trials with high-probability cues are included. (**C, D**) Responses of the neurons (**D**) color-coded in **C** during five sample trials (rows) for four different cue-outcome conditions (columns). Dotted lines indicate the tone and reward onsets. Scale bar in **C** represents 100 μm. Lines above each plot show when licks occurred. (**E**) Percentage of task-modulated neurons that were selectively active during cue, post-reward licking, or post-licking epochs of the task. Error bars represent 95% confidence intervals. (**F**) Population-averaged responses of task-modulated neurons selectively active during the three epochs. Data for neurons active during the post-reward licking period are separately shown aligned to the first and the last lick. (**G**) Session-averaged activity of all task-modulated neurons (left) and those that were significantly active during only one of three task epochs (right). Neurons were sorted by the timing of their peak activity.

DOI: https://doi.org/10.7554/eLife.32353.009

The following figure supplement is available for figure 4:

**Figure supplement 1.** Temporal specificity of post-reward licking responses.

DOI: https://doi.org/10.7554/eLife.32353.010

matrix exhibited qualitatively similar responses, but there was a significantly stronger tone-evoked activation in striosomes than in the nearby matrix regions sampled (*Figure 5A*) (ANOVA main effect p<0.001). Moreover, the high-probability tone cue evoked a larger response than the low-probability tone cue (p<0.001), and there was a trend for an interaction between compartment and tone (p=0.055).

We tested for the selectivity of the responses to the high- and low-probability tone cues by quantifying the area under the receiver operating characteristic curve (AUROC) for these responses. Both striosomal and matrix responses displayed significant selectivity for the high-probability tone (p<0.05), but there was no difference in selectivity between the striosomes and matrix at this stage of learning. We also performed an AUROC analysis to estimate the selectivity for rewarded trials (*Figure 5B*). Both striosomal and matrix neuropil had elevated activity in rewarded trials, compared to non-rewarded trials, with both high- and low-probability tones. Repeated measures ANOVA showed that striosomes had a higher selectivity for rewarded trials than did the matrix (ANOVA main effect p<0.001). The selectivity for reward was larger in low-probability tone trials than in high-probability tone trials for both compartments (ANOVA main effect p<0.001), but there was no interaction between cell-type and selectivity for reward (*Figure 5B*). Thus, both striosomes and matrix were more activated when the reward was less expected. We also tested how the beginning and end of licking were reflected by activity in the two compartments (*Figure 5C*). The striosomal activation was higher than matrix activation (ANOVA main effect p<0.001), and the activation was larger at the end of licking than at the beginning (*Figure 5C*) (±1 s around event, ANOVA main effect p<0.001). Thus, the overall fluorescence shows that both striosomes and matrix are active during the trials, but that striosomes are more strongly activated, particularly during the cue period, and also at the end of the post-reward licking period. However, we note that summing activity over populations of neurons makes it impossible to dissociate the reward-related activation from the carry-over effects of the tone-related activation.

Next, we analyzed single-cell calcium responses of striosomal and matrix neurons during the task. Individual neurons in both compartments were active during the task (*Figure 5D*). We found a higher proportion of striosomal neurons (42.9%; 312 out of 712 neurons) than matrix neurons (36.5%; 721 out of 1977 neurons) that were task-modulated (p<0.005, Fisher's exact test; *Figure 5E*). Among the task-modulated neurons, a higher percentage of striosomal neurons was active during the cue epoch of the task (23.7% of striosomal, 13.7% of matrix, p<0.001, Fisher's exact test). By contrast, we found no differences in the percentages of striosomal and matrix neurons that were active during the post-reward licking or post-licking period (p>0.05). In plots of session-averaged activity sorted by the timing of peak responses, we observed that striosomal and matrix neuron activities similarly spanned each of the three task epochs, as though they tiled the temporal space of the task (*Figure 5F*). We found no differences in the trial-to-trial reliability of striosomal and matrix responses during the task epochs (*Figure 5—figure supplement 1*). To compare responses of all task-modulated striosomal and matrix neurons during these epochs, we analyzed population-averaged activity aligned to different task events (*Figure 5G–J*). As in our neuropil analyses, we found that individual striosomal neurons were more robustly active than individual matrix neurons during the cue epoch of the task (*Figure 5G*, ANOVA main effect p<0.001). Moreover, the high-probability tone elicited a higher response than the low-probability tone (p<0.001).

We used an AUROC analysis to compare activity in trials that were rewarded (responses aligned to first lick after reward delivery) or unrewarded (responses aligned to 2 s after cue onset, a time period matching that for the rewarded trial analysis). We found that striosomal neurons were more selective for rewarded trials (*Figure 5H*, p<0.001). The selectivity for reward was greater for low-probability than for high-probability tone trials (p<0.01). Although neurons in the two compartments responded similarly during post-reward licking (*Figure 5I*, p>0.05), striosomes had a higher response during the post-licking period (*Figure 5J*, p<0.01). Together, these findings demonstrate that neurons in both the striosome and matrix compartments are task-modulated in relatively similar patterns in this appetitive classical conditioning task, that neurons in striosomes are more strongly task-modulated than neurons in the nearby matrix, and that they are particularly more active during reward-predicting cues.

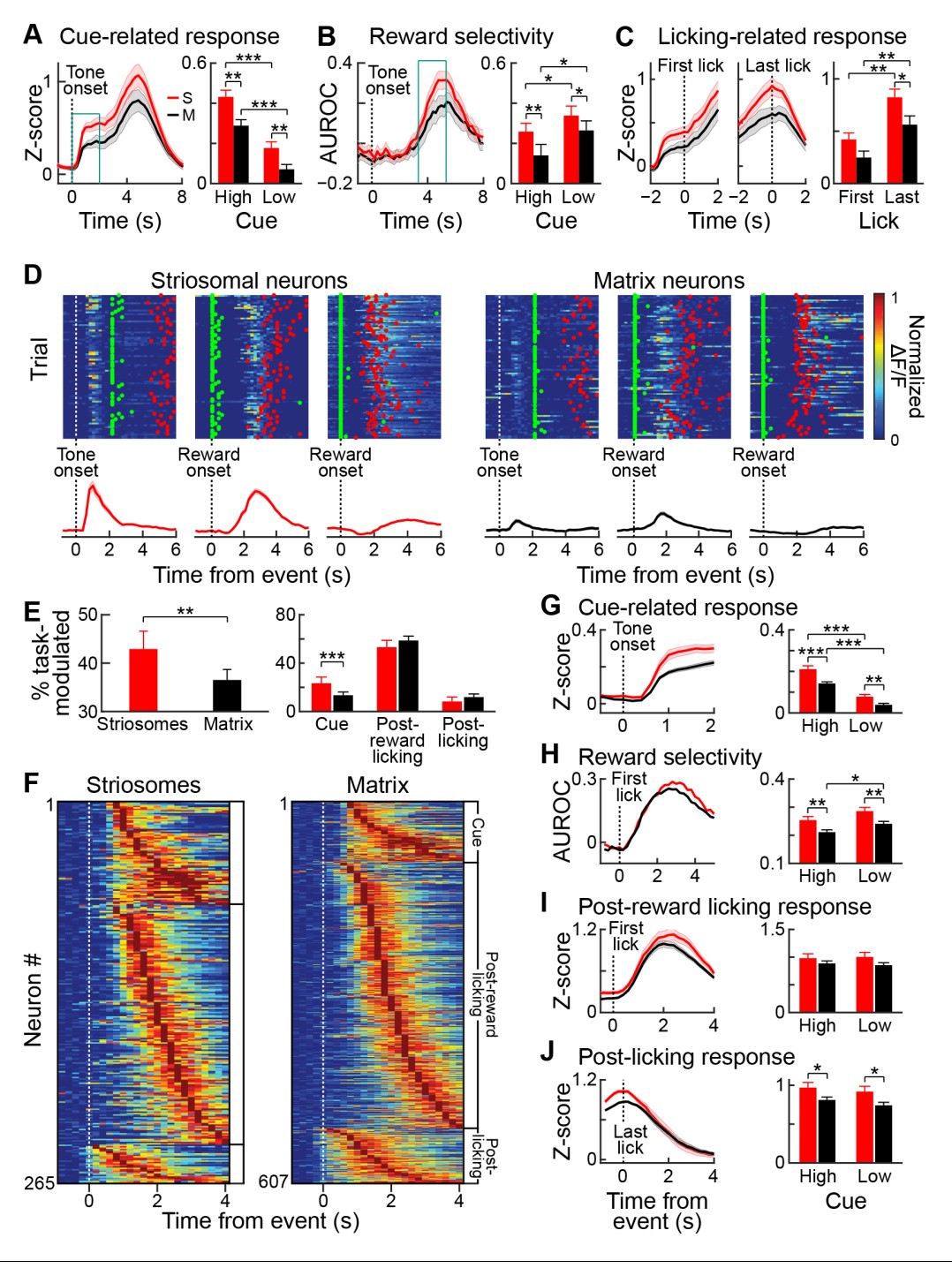

**Figure 5.** Striosomal neurons respond more strongly to reward predicting cues than matrix neurons. (**A**) Average striosomal (S, red) and matrix (M, black) neuropil activation during rewarded trials with high-probability cue (left), and quantification of the magnitude of the response to high- and low-probability cues (right), calculated for the time period indicated by blue box (left). **p<0.01, ***p<0.001 (ANOVA and post hoc t-test). Shading and error bars represent SEM. (**B**) Neuropil selectivity for the rewarded vs. unrewarded trials for every time point in the trials in high-probability trials (left) and the average selectivity during the time indicated in the blue box (left) for both trial types (right). (**C**) Average neuropil post-reward activity aligned to the first (left) or last lick (middle), and average response during the ±1 s period (right). *p<0.05, **p<0.01 (ANOVA and post hoc t-test). (**D**) Trial-by-trial response of three striosomal (left block) and three matrix (right block) neurons that were selectively active during the cue (left), post-reward licking (middle), or end of licking (right) task-epochs. Green and red dots show,

*Figure 5 continued on next page*

*Figure 5 continued*

respectively, the first lick after reward delivery and the last lick. Average responses for the same neurons are shown underneath the color plots. (**E**) Proportion of all task-modulated striosomal and matrix neurons (left) and those that were modulated selectively during cue, post-reward licking, or post-licking epochs of the task (right). \*\*p<0.01, \*\*\*p<0.001 (Fisher's exact test). Error bars represent 95% confidence intervals. (**F**) Session-averaged responses of all task-modulated striosomal (left) and matrix (right) neurons, plotted on the color scale shown in D. Neurons are grouped and sorted as were those shown in *Figure 4G*. (**G**) Population-averaged responses of all task-modulated striosomal and matrix neurons to the high-probability cue (left), and the population responses separately averaged for high- and low-probability cues (right). \*\*p<0.01, \*\*\*p<0.001 (ANOVA and post hoc t-test). Shading and error bars represent SEM. (**H**) Discriminability between rewarded and unrewarded trials for striosomal and matrix neurons. Left plot shows selectivity during trials with high-probability cue, and right plot shows average discriminability for all trials (quantified over 1–2 s time window after reward delivery). \*p<0.05, \*\*p<0.01 (ANOVA and post hoc t-test). (**I,J**) Population-averaged response during post-reward licking (**I**) or post-licking (**J**) periods, with data aligned, respectively, to first and last lick after reward delivery. \*p<0.05 (ANOVA and post hoc t-test).
DOI: https://doi.org/10.7554/eLife.32353.011

The following figure supplement is available for figure 5:

**Figure supplement 1.** Response reliability of task-related responses of striosomal (red) and matrix (black) neurons.
DOI: https://doi.org/10.7554/eLife.32353.012

## Striosomal tone-evoked responses are acquired during learning

To determine how these responses were shaped by training, we analyzed striatal activity during the acquisition period of the task. To quantify levels of learning, we tested for significance in the difference between anticipatory licking for the high- and low-probability cues during the tone presentation and the reward delay. If mice exhibited a significant difference on 2 out of 3 consecutive days, we considered them as being trained. Sessions performed before this criterion was met were categorized as acquisition sessions. This categorization allowed us to ask whether the strong striosomal cue-related response was a sensory feature, or whether it was an acquired response related to the meaning of the stimulus. Of the five mice studied, two were initially trained on a three-tone version of this task and were therefore excluded from the analysis of the initial training period (*Table 4*). The three mice included and the two mice excluded from the training data set had similar baseline ΔF/F values and percentages of task-modulated and tone-modulated neurons. Activity measures for the neuropil signals during training for all sessions before the mice reached the learning criterion (n = 33) were compared with the signals in the sessions after criterial performance had been met (n = 20). The striosomal responses to the tones were much stronger after animals learned the task (*Figure 6A*). The neuropil signal in striosomes was significantly higher after the task performance

**Table 4.** Data details for individual mice.

| | Mouse | | | | |
|---|---|---|---|---|---|
| | **1** | **2** | **3** | **4** | **5** |
| Number of acquisition sessions | 12 | 11 | 10 | 19 + 2 * | 12 + 4 * |
| Number of criterion sessions | 7 | 9 | 4 | 5 | 8 |
| Number of overtraining sessions | 5 | 5 | 0 | 0 | 0 |
| Mean baseline ΔF/F | 11.2 (0.7) | 10.8 (0.7) | 12.4 (1.4) | 12.1 (1.0) | 12.7 (0.6) |
| Standard deviation baseline ΔF/F | 35.5 (1.3) | 41.3 (1.3) | 30.6 (1.6) | 37.9 (1.7) | 39.8 (1.2) |
| Maximum baseline ΔF/F | 237.2 (9.6) | 296.3 (11.2) | 148.8 (9.3) | 236.1 (13.2) | 270.3 (9.7) |
| Task-modulated neurons in striosomes | 54.1% | 22.4% | 31.7% | 66.2% | 46.3% |
| Task-modulated neurons in matrix | 46.9% | 20.8% | 19.4% | 56.4% | 40.6% |
| Tone-modulated neurons in striosomes | 21.6% | 14.0% | 14.6% | 16.9% | 7.3% |
| Tone-modulated neurons in matrix | 13.0% | 9.2% | 3.3% | 9.5% | 4.4% |

*Two mice were initially trained on a more complex version of the task with three tones instead of two (numbers of sessions trained on the two versions indicated by the first and second number, respectively). Data of these mice were excluded from acquisition analyses.
DOI: https://doi.org/10.7554/eLife.32353.014

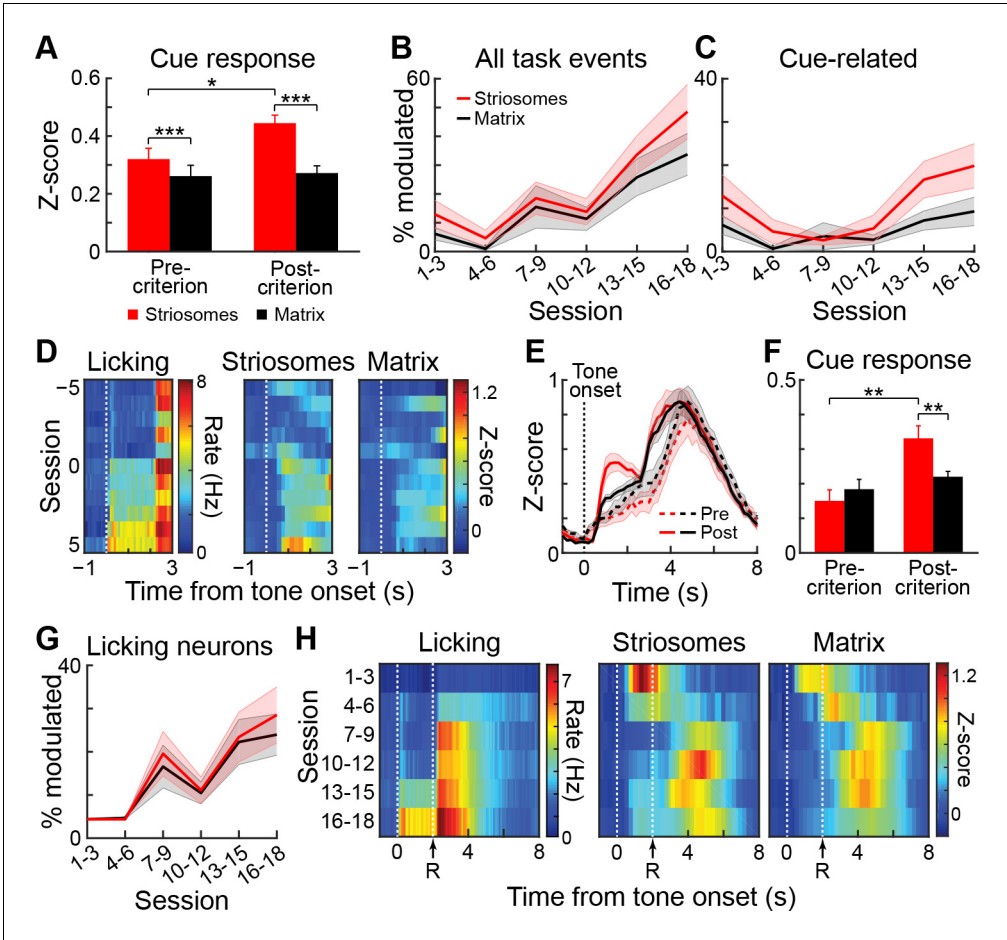

**Figure 6.** Cue-related signals in striosomes develop during training. (**A**) Average total striosomal (red) and matrix (black) neuropil signal during the 1.5 s tone period in all sessions before and after reaching learning criterion. *p<0.05, ***p<0.001 (ANOVA and post hoc t-test). Error bars represent SEM. (**B, C**) Percentage of task-modulated (**B**) and cue-modulated (**C**) neurons in striosomes and matrix during the course of training. Shading represents SEM. (**D**) Mean normalized (z-score) licking (left) and ΔF/F activity in task-modulated striosomal and matrix neurons (right) during ±5 sessions around the session in which the learning criterion was reached (session 0). (**E**) Activity of task-modulated striosomal and matrix neurons averaged and normalized for blocks of 5 sessions before (dotted lines) and after (solid lines) the learning criterion was reached. (**F**) Quantification of the mean response of all task-modulated neurons during the period from tone onset to reward onset. **p<0.01 (ANOVA and post hoc t-test). (**G**) Percentage of neurons modulated in the licking period during training. (**H**) Mean normalized (z-score) licking (left) and activity of task-modulated neurons (right) during training around the time of reward delivery (R).
DOI: https://doi.org/10.7554/eLife.32353.013

reached the training criterion than before this point (p<0.05). Such a difference was not observed for the matrix (p>0.05; ANOVA interaction p<0.001). In order to perform a similar analysis for single cells, we grouped responses in consecutive three-day bins. The results indicated that during training, the percentage of task-modulated neurons increased steadily (**Figure 6B**), but that when mice reached the learning criterion (sessions 11 and 12 for the mice shown in **Figure 6B,C**), there was a rapid increase in the proportion of cue-modulated neurons (**Figure 6C**), most notably among striosomal neurons.

We further tested whether there was a sudden step-like increase in striosomal tone signaling during training. We averaged the z-scores of the activity of all task-modulated neurons for each of the last five sessions before criterial performance, for the session in which the learning criterion was met, and for each of the first five sessions after the criterial session (**Figure 6D**). Comparing these values indicated a clear increase in striosomal signaling during the tone (**Figure 6E**) when the mice began

to exhibit differential licking responses to the two tones. This increase in striosomal activity was significant (*Figure 6F*, ANOVA, training main effect p<0.001; interaction p<0.05). In addition to this development of tone responsiveness, there was a tone-related activation in striosomes in the sessions in which the animals were first exposed to the task (*Figure 6C*), perhaps reflecting a surprise or novelty signal effect. This tone-related activation disappeared after 1–3 sessions and then reemerged later as mice learned the task. There was also an increase in the percentage of neurons that responded during the post-reward licking period (*Figure 6G*), and the average activity of all task-modulated neurons during training increased in the period after reward delivery (*Figure 6H*). In contrast to the increases in tone response, this reward-period increase occurred several sessions before mice learned the task.

## During overtraining, tone-related responses of striosomal neurons intensify and become increasingly selective for high-probability tones

To investigate further the relationship between neuronal responses and learning, two mice were trained for an additional five sessions. In these overtraining sessions, we imaged again the same fields of view from which we had collected movies during the criterion phase (*Figure 7*). The tone-evoked aggregate response became notably higher and sharper during this phase (*Figure 7A*). The increase in responses related to the tone during overtraining was particularly strong in striosomes. By contrast, the reward period activation immediately following the peak of the cue-evoked response was reduced. The signal initially dropped compared to the earlier sessions but subsequently reached the same magnitude, thus resembling previously reported task-bracketing patterns (*Barnes et al., 2005*; *Jin and Costa, 2010*; *Jog et al., 1999*; *Smith and Graybiel, 2013*; *Thorn et al., 2010*). This pattern contrasted with the cue-evoked licking response (*Figure 7B*), which remained high after the high-probability cue, when the bracketing-like effect in the ΔF/F signal was greatest. We compared the peak responses of the striosomal and matrix samples during the tone presentation period for the acquisition, post-criterion and overtraining sessions (*Figure 7C*), and found a highly significant interaction (ANOVA interaction p<0.005). In the trained and overtrained mice, striosomes had significantly higher tone-evoked responses than did the matrix (paired t-test, trained mice p<0.01 and overtrained mice p<0.05). The striosomal neuropil responses also became more selective for the high-probability cue during overtraining (*Figure 7D*), so that during overtraining the striosomal selectivity was significantly larger than the matrix response (paired t-test p<0.05).

The percentage of task-modulated neurons grew with training, then slightly dropped during overtraining (*Figure 7E*, left; striosomes: 12.2% during acquisition, 42.9% after criterion and 37.7% during overtraining; matrix: 8.6% during acquisition, 36.6% after criterion and 27.5% during overtraining). At all stages, there were a higher percentage of striosomal task-modulated neurons than matrix neurons responding in the task (Fisher's exact test, p<0.01). By contrast, the percentage of cue-modulated neurons (*Figure 7E*, second panel) grew further during overtraining (striosomes: 4.1% during acquisition, 15.0% after criterion and 21.1% during overtraining; matrix: 2.3% during acquisition, 8.1% after criterion and 13.9% during overtraining). There were more tone-modulated neurons in striosomes than in matrix during acquisition, after criterion and during overtraining (Fisher exact test, p<0.05). The percentage of cells that were active during the post-reward licking period (*Figure 7E*, third panel) increased during training but went down during overtraining (striosomes: 7.0% during acquisition, 28.9% after criterion and 14.9% during overtraining; matrix: 5.6% during acquisition, 27.0% after criterion and 13.3% during overtraining), but there were no differences between striosomes and matrix (Fisher's exact test, p>0.05). The proportion of neurons activated after the end of licking remained stable for both striosomal and matrix neurons during overtraining (*Figure 7E*, fourth panel; striosomes: 1.9% during acquisition, 5.6% after criterion and 6.6% during overtraining; matrix: 1.1% during acquisition, 7.3% after criterion and 6.8% during overtraining). We found no differences between striosomes and matrix at any training stage (Fisher's exact test, p>0.05).

The limited number of significantly modulated neurons in these two mice was too small to make further statistical comparisons between the neuronal responses. Nevertheless, the findings for the entire performance period of the mice collectively demonstrate that the activity patterns observed after training were largely acquired during training, that the strengthening of the tone response was greater for striosomes than for matrix, that this response emerged at the time the animals began differentially responding to the tones, and that this response developed further during overtraining,

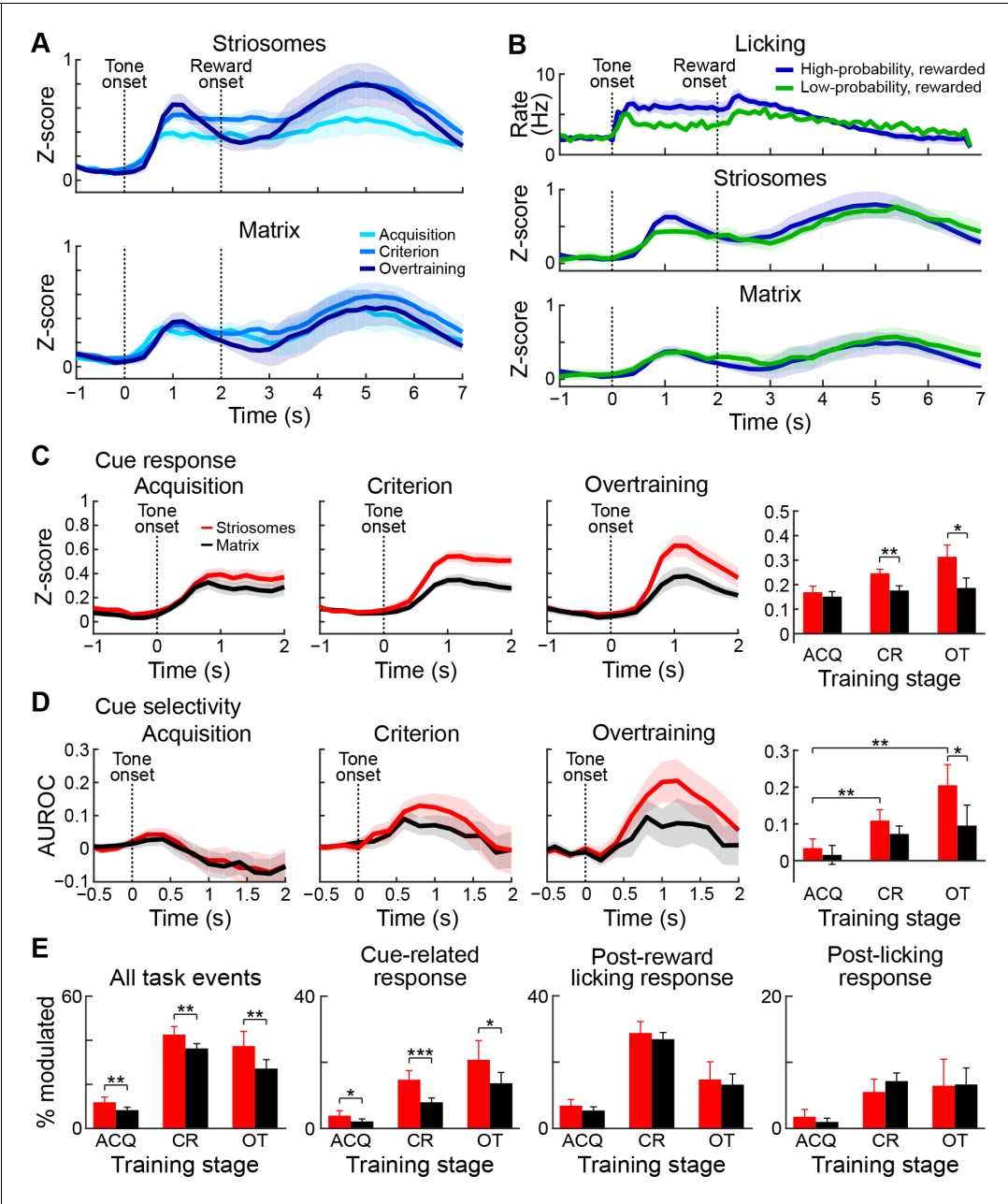

**Figure 7.** Striosomal cue-related responses strengthen during overtraining and become more selective. (**A**) Mean neuropil signals during acquisition (light blue), after learning criterion (medium blue) and during overtraining (dark blue) in striosomes (top) and matrix (bottom). Shading represents SEM. (**B**) Average licking (top) and neuronal activity in striosomes (middle) and matrix (bottom) in rewarded trials with high- (blue) and low- (green) probability cues. (**C**) Mean neuropil responses in striosomes (red) and matrix (black) during acquisition, after criterial performance and during overtraining, and the mean of the response sizes (right). *p<0.05, **p<0.01 (ANOVA). Shading and error bars represent SEM. (**D**) Selectivity for the high-probability cue, shown as in B. *p<0.05, **p<0.01 (ANOVA). (**E**) Percentages, shown in panels from left to right, of task-modulated neurons (left), tone-modulated neurons (second), neurons modulated in post-reward licking period (third) and neurons modulated in post-licking period (right) during acquisition (ACQ), after criterion (CR) and during overtraining (OT). *p<0.05, **p<0.01, ***p<0.001 (Fisher's exact test).
DOI: https://doi.org/10.7554/eLife.32353.015

The following figure supplement is available for figure 7:

**Figure supplement 1.** The size of tone-evoked ΔF/F activation increases as behavioral performance improves, particularly as seen in the calcium activity in striosomes.
DOI: https://doi.org/10.7554/eLife.32353.016

becoming larger and more selective for the high-probability cue. Because the definition of the training phases was by necessity somewhat arbitrary, and the behavioral performance of the mice could fluctuate across days, we used linear regression to test how well the behavioral performance could predict the ΔF/F activation in striosomes and matrix. For every session, we calculated the mean and standard deviation of the baseline period (1 s preceding the cue onset) and then calculated tone-evoked licking and ΔF/F responses of the neuropil signal in z-scores. We found that in sessions in which the tone-evoked licking was greater, the neuropil response was also greater (*Figure 7—figure supplement 1*). To test this relationship, we first made two separate models for striosomes and matrix. The regression coefficients for licking were significant for both compartments for the high-probability cue responses but not for the low-probability cue responses (*Table 5*). When we tested how well the difference in licking during both cue periods could predict the difference in ΔF/F activation during the two cue types, we found a significant regression coefficient for striosomes, but only a trend for the matrix. Next, we made a combined model accounting for ΔF/F activation as a function of licking and quantified the residuals for striosomes and matrix. For both cues and for the difference between them, we found that the striosomal residuals were significantly bigger than those for the matrix. Together, these linear regression analyses demonstrate that in sessions in which the behavioral performance was better, the neuronal response was larger, especially in striosomes. Thus, the behavior was predictive of the neural response, particularly for striosomes.

## Matrix responses are more sensitive to recent outcome history than are striosomal responses

In the classical conditioning task employed in this study, mice used the auditory tone presented during the cue epoch to guide their expectation for receiving a reward in the current trial. We examined their licking responses as a proxy for such expectation in order to ask whether, in addition to the information provided by the cue, the mice used the outcome of the previous trial to tailor their reward expectation in the current trial. In trials following rewarded trials, mice showed increased anticipatory licking during the cue and reward delay (*Figure 8A*, left and right; n = 33 sessions from five mice; p<0.001, Wilcoxon signed-rank test), but licking during the post-reward period was unaffected by outcome in the previous trial (*Figure 8A*, middle and right; p>0.05, Wilcoxon signed-rank test).

To determine whether the task-related activity of the striatal neurons in our sample was also modulated by outcome history, we compared the activities in trials preceded by a rewarded or unrewarded trial, regardless of the cue type (high- or low-probability) presented in the current trial. We first analyzed the effect of reward history on the cue-period responses of single task-modulated neurons and found that activity was slightly greater when the previous trial was rewarded (mean z-scores: 0.21 ± 0.01 vs. 0.17 ± 0.01 for previously rewarded and unrewarded; p<0.01). However, when we analyzed the effect of outcome history on neural responses observed during post-reward licking in currently rewarded trials, we found that the activity of a subset of striatal neurons was highly sensitive to outcome in previous trials (*Figure 8B*). Activity during post-reward licking was

**Table 5.** Outcome of the regression analyses.

| | | Trial type | | |
| --- | --- | --- | --- | --- |
| | | **High-probability cue** | **Low-probability cue** | **Difference (high − low)** |
| Striosome model | Regression coefficient | 0.069 *** | 0.057 | 0.074 ** |
| | R-squared | 0.180 | 0.029 | 0.095 |
| Matrix model | Regression coefficient | 0.042 * | 0.016 | 0.056 |
| | R-squared | 0.081 | 0.003 | 0.053 |
| Combined model | Regression coefficient | 0.056 *** | 0.037 | 0.065 ** |
| | R-squared | 0.118 | 0.014 | 0.072 |
| | Residual for striosomes (mean ± SEM) | 0.053 ± 0.021 *** | 0.031 ± 0.021 *** | 0.022 ± 0.023 *** |
| | Residual for matrix (mean ± SEM) | −0.053 ± 0.021 *** | −0.031 ± 0.018 *** | −0.022 ± 0.024 *** |

*p<0.05, **p<0.01, ***p<0.001

DOI: https://doi.org/10.7554/eLife.32353.017

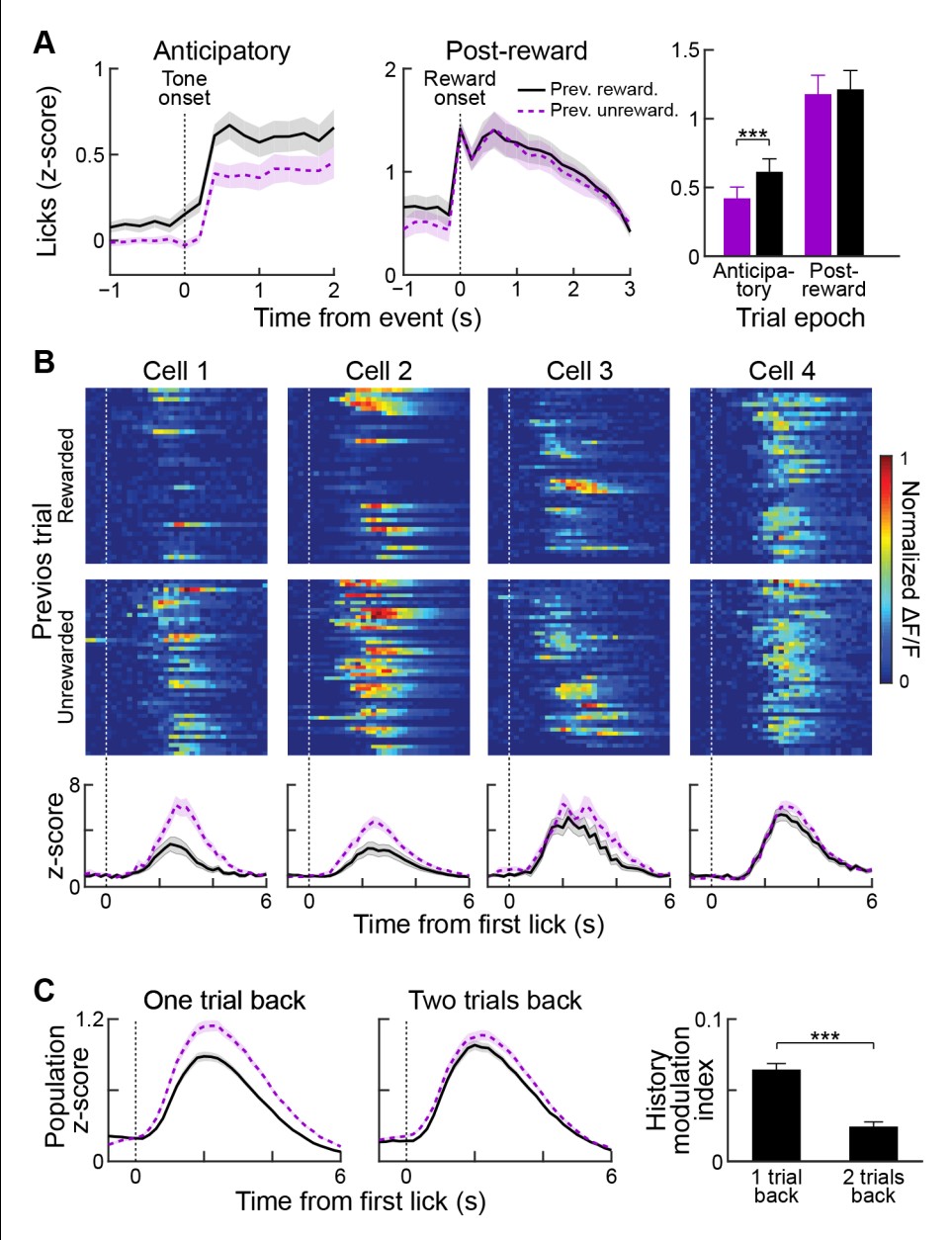

**Figure 8.** Reward history modulates anticipatory licking behavior and licking-period responses in striatal neurons. (**A**) Session-averaged licking activity during anticipatory and post-reward periods for trials in which the previous trial was rewarded (black solid lines) or unrewarded (purple dotted lines). Bar plot (right) shows modulation of anticipatory or post-reward licking activity by reward history. ***p<0.001 (Wilcoxon signed-rank test). Shading and error bars represent SEM. (**B**) Single-trial (top two rows) and averaged (bottom row) post-reward licking responses of four sample neurons for previously rewarded (black solid) or unrewarded (purple dotted) trials. (**C**) Population-averaged post-reward licking activity of all task-modulated neurons for reward histories extending one or two trials back. ***p<0.001 (Wilcoxon signed-rank test).
DOI: https://doi.org/10.7554/eLife.32353.018

enhanced when the previous trial was unrewarded, compared to when the previous trial was rewarded. Similarly, population-averaged responses of task-modulated neurons were significantly higher when the previous trial was unrewarded, as compared to when it was rewarded (p<0.001, Wilcoxon rank-sum test). Importantly, post-reward licking behavior was invariant to previous trial

outcome (*Figure 8A*), making it unlikely that the observed changes in neural activity were related to changes in the motor output during reward consumption.

To determine how far back in time we could detect an outcome history effect, we computed a history modulation index (see Materials and methods) for currently rewarded trials with two types of reward history. In the first group, we separated rewarded trials based on whether the previous trial was rewarded or unrewarded (one trial back). For the second group, we disregarded the outcome status in the immediately preceding trial and separated trials depending on the outcome status of two trials in the past (two trials back). This analysis showed that recent reward history has a stronger influence on post-reward licking responses of task-modulated neurons than trials farther back in the past (*Figure 8C*, p<0.001, Wilcoxon signed-rank test).

We asked whether this history modulation effect was detectable for both striosomal and matrix neurons (*Figure 9*). Examination of both population-averaged responses (*Figure 9A*) and single-cell responses (*Figure 9B*) suggested that both striosomal and matrix neurons were modulated by previous reward history, but that matrix neurons were more sensitive to this modulation. Quantification of this comparison by calculating the history modulation indices for striosomal and matrix neurons confirmed that the matrix responses were more influenced by previous reward history than the responses of striosomal neurons (*Figure 9C,D*; p<0.01, Wilcoxon rank-sum test).

## Discussion

Our findings demonstrate that 2-photon calcium imaging can be used to identify the activity patterns of subpopulations of neurons distinguished as being in either the striosome or matrix compartments of the striatum. Even with the use of a simple classical conditioning task involving cues predicting high or low probabilities of receiving reward, we could detect in all mice many task-related striatal neurons, altogether 38% of the 2704 neurons successfully imaged in the post-training phase. We found a remarkable parallel in many of the responses of neurons in striosomes and neurons in the matrix. Yet we also found clear differences in the responses of the striosomal and matrix neurons during cue presentation, found contrasts in the timing and selectivity of striosomal and matrix responses during learning and overtraining, and found that the responses of the two compartments were differentially affected by reward history. These findings, based on direct visual detection of striosomes by their birthdate-labeled neuropil and cell bodies, demonstrate that neurons of the two main compartments of the striatum, even though sharing many basic features of neuronal responses during reward-based conditioning, have distinguishable response properties that hint at distinct encoding functions of striosomal and matrix neurons related to reinforcement learning and performance. These findings suggest that in vivo imaging of striosomes and matrix could succeed in solving long-standing questions about the functions of these two major compartments of the striatum.

### Tone-period activity

By the time the animals had reached the learning criterion, neurons in both compartments had developed task-related responses, and striosomes, examined both by averaged neuropil measures and by single-cell activities, were more responsive to the task than were neurons in the surrounding matrix. The differential activation of striosomes was particularly striking for responses to the reward-predictive cues. More neurons of the striosomal population were active in relation to the cues, and this effect grew stronger as animals acquired the task. The striosomal neurons also were more selective for the high-probability cue than were the matrix neurons. Greater responsivity to predictors of reward in the response profiles of striosomal cells is in line with striosomes acting as critic in an actor-critic architecture (*Doya, 1999*). However, our results are also in accord with other ideas based on limbic associations of the striosomes (*Amemori et al., 2011*). The enhanced striosomal responses to cues did not reflect an overall greater response of striosomes to all conditions; for example, their responses were less sensitive than those of the matrix neurons to immediate reward history. Thus, striosomal neurons stood out as more sensitive to the cues indicating reward.

### Outcome period activity

Over the task-related population, the highest activity levels for many of the neurons as the learning criterion was reached occurred during the outcome period, whether the neurons were in striosomes

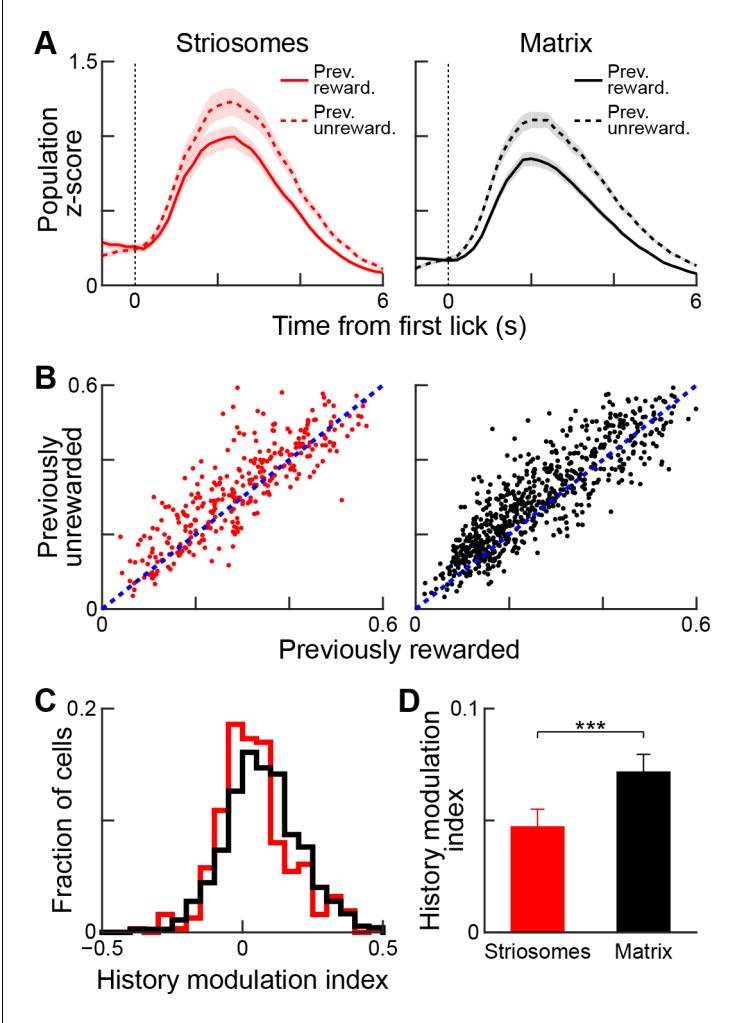

**Figure 9.** Reward-history modulation of striosomal and matrix neurons. (**A**) Population-averaged responses of all task-modulated striosomal (red, left) and matrix (black, right) neurons during trials following previously rewarded (solid) or unrewarded (dotted) trials. Shading represents SEM. (**B**) Normalized lick-period responses (averaged over 1–3 s after first lick) of individual striosomal (left) and matrix (right) neurons. Responses with previously rewarded trials (x-axis) are plotted against responses from previously unrewarded trials (y-axis). Unity line is shown as blue dotted line (**C**) Histogram showing reward-history modulation index for all task-modulated striosomal and matrix neurons. (**D**) Mean reward-history modulation index for striosomal and matrix neurons. ***p<0.001 (Wilcoxon rank-sum test). Error bars represent SEM.

DOI: https://doi.org/10.7554/eLife.32353.019

or in the surrounding matrix. Thus, the compartments seemed equivalently engaged: the highest percentages of neurons of both types were active during this period. Neuronal activity built up and peaked at the end of the licking, leading to the obvious possibility that this activity was primarily related to licking itself. However, several factors pointed to this response as being different from a pure motor response related to the licking movements. Most strikingly, even among the neurons strongly active during the prolonged licking period, the majority rose to their peak activity at specific times within this period rather than during the entire licking period. These post-reward peak responses, collectively, appeared to cover the entire time after reward. A subgroup of these neurons even peaked in activity after the end of the last licks, resembling neuronal activity in electrophysiological recordings (*Barnes et al., 2005*; *Jin and Costa, 2010*; *Jog et al., 1999*; *Smith and Graybiel, 2013*; *Thorn et al., 2010*). Second, we found dissociations between licking behavior and neuronal responses. For instance, activations during licking were larger when the previous trial was

unrewarded than when it was rewarded, whereas the licking behavior itself was not different. The anticipatory licking during the cue period and the neuronal responses during the cues also appeared dissociable during overtraining. During high-probability cues, when animals licked throughout the reward delay period, the neuronal signal decayed, whereas the opposite occurred during low-probability cues. Thus, although the signals observed during periods of licking were likely to be related to licking, their patterns of occurrence suggest an interesting multiplexing of information about licking, reward prediction, timing with respect to task events, and reward history. Finally, the differences in activity in striosomes and matrix that we observed cannot be accounted for by differences in licking behavior during the imaging of these compartments, because the effects were also visible when analyzing neuropil activity, in which case matched, simultaneously registered striosomal and matrix data points were acquired from every session during the same behavioral performance.

## Sensitivity to reward history

In contrast to these accentuated responses of striosomes, striosomal neurons as a population were relatively less sensitive than those in the matrix to immediate reward history, although again, both populations were modulated in parallel so that the differences were quantitative, not qualitative. When the learning criterion had been reached, the neuronal responses for a given trial were elevated when the previous trial was not rewarded. By contrast, anticipatory licking was decreased in trials following unrewarded trials. These effects were significantly larger for the matrix. This reward history effect was much smaller for two-back reward history, suggesting that it reflected immediate reward history. Given the limits of our data set, we could not determine the mechanism underlying this difference in sensitivity to reward history.

## Learning-related differences in the responsiveness of striosomal and matrix neurons

Our recordings during the course of training demonstrated that both the cue-related responses and the post-reward responses were built up in striosomes and nearby matrix regions during behavioral acquisition of the task, with tone-related responses abruptly appearing when the mice reached the learning criterion. These learning-related dynamics suggest that the observed tone responses do not simply reflect responses to auditory stimulus presentations.

During overtraining, the striosomal cue response strengthened: more striosomal neurons were significantly modulated by tone presentation, this striosomal response became stronger and more temporally precise, and it became more selective for the high-probability cue. By contrast, the activity in the period after reward delivery until the end of licking did not change notably and was even reduced slightly but non-significantly. Finally, the overall activity patterns in the neuropil began to resemble the classical task-bracketing pattern with peaks of activity at the beginning and the end of the trial (*Barnes et al., 2005*; *Jin and Costa, 2010*; *Jog et al., 1999*; *Smith and Graybiel, 2013*; *Thorn et al., 2010*).

In the matrix, the effects of overtraining were less pronounced. The responses to the tone and reward consumption remained similar, but, as in the striosomes, a pattern resembling task-bracketing formed in the matrix. All of these effects could be detected not only at the single-cell level but also by assessing total fluorescence in defined striosomes and regions of the nearby matrix with equivalent areas. These findings suggest that although both compartments have cue-related responses, in striosomes the responses to reward-predicting cues are accentuated relative to responses detected in the matrix and are particularly increased with extended training.

## Reward signaling in the dorsal striatum

It has previously been found that a minority of dorsal striatal neurons encode reward prediction errors (*Oyama et al., 2010*, *2015*; *Stalnaker et al., 2012*). Two of the major targets of striosomes, the dopamine-containing substantial nigra pars compacta and, via the pallidum, the lateral nucleus of the habenula, are well known to signal reward prediction errors (*Bayer and Glimcher, 2005*; *Bromberg-Martin and Hikosaka, 2011*; *Keiflin and Janak, 2015*; *Matsumoto and Hikosaka, 2007*; *Schultz, 2016*; *Schultz et al., 1997*). Therefore, we asked whether striosomes and matrix differentially encode reward prediction error signals. One particular possibility is that striosomes through their GABAergic innervation of dopamine-containing neurons could transmit a negative reward

prediction signal. We found that striosomes preferentially encoded reward-predictive cues. We did not find significant differences between striosomes and matrix in outcome-related activity. We also did not find prominent signals related to reward omissions in either striosomes or matrix. Some models of striosome function posit that striosomes would have such signals. Our task, however, was a simple one and likely did not draw out such activity, and we did not have a full data set for the overtraining period, when such responses might be predicted to become apparent. We also note that we were unable to test hypotheses suggesting that tasks with multiple contexts and decision-making modes could be important for striosomal activation. Finally, we did not address motivational conflict, stress or anxiety states as potentially being critical to striosomal activation (*Amemori and Graybiel, 2012*; *Friedman et al., 2017*, *2015*).

We are also aware that the dorsal striatum is heavily implicated in motor behavior, through learning, action selection and perhaps the invigoration of action (*Amemori et al., 2011*; *Apicella et al., 1992*; *Balleine et al., 2007*; *Cui et al., 2013*; *Hikosaka et al., 2014*; *Howe et al., 2013*; *Klaus et al., 2017*; *Kreitzer and Malenka, 2008*; *Mink, 1996*; *Nelson and Kreitzer, 2014*; *Niv et al., 2007*; *Packard and Knowlton, 2002*; *Redgrave et al., 1999*; *Salamone and Correa, 2012*; *Samejima et al., 2005*; *Yin and Knowlton, 2006*). Nevertheless, we chose to start in these experiments by determining how fundamental features of the striatum, signaling of outcome and prediction of outcome, are represented in the responses of neurons in the striosome and matrix compartments. Future work will address the involvement of striosomes and matrix in action and decision-making among alternative options.

## Striosome labeling

Visual identification of striosomes by their dense neuropil labeling was achieved by pulse-labeling of striosomal neurons and their processes at the mid-point of striosome neurogenesis. Even though minorities of the striosomal neurons were pulse-labeled by the single tamoxifen injections, and despite the fact that there were scattered birthdate-labeled neurons in the extra-striosomal matrix at the striatal levels examined (ca. 15% of tdTomato-positive neurons), we could readily identify striosomes visually in vivo using 2-photon microscopy and could confirm this identification in post-mortem MOR1-counterstained sections prepared to assess the selectivity of labeling. We are aware that, with the use of pulse-labeling at neurogenic time points, we have incomplete labeling of compartments in any one animal, but the time of induction that we used was at the middle of the striosomal neurogenic window and was before the onset of major levels of matrix neuron neurogenesis in the striatal regions imaged (*Fishell and van der Kooy, 1987*; *Graybiel, 1984*; *Graybiel and Hickey, 1982*; *Hagimoto et al., 2017*; *Kelly et al., 2017*; *Newman et al., 2015*). We are also aware that the matrix compartment itself is heterogeneous, as it is composed of many input-output matrisome modules (e.g., *Eblen and Graybiel, 1995*; *Flaherty and Graybiel, 1994*), but such heterogeneity could not be taken into account in our experiments. We did choose for analysis zones in the matrix that were close to the striosomes studied. Our method did not rely on a single molecular or genetic marker to distinguish compartmental identify, but this feature had also a possible advantage in thereby avoiding potential unidentified biases that could arise from molecular-identity labeling.

It is currently unknown to what extent there are different subtypes of striosomal neurons and what the exact neuronal subtype composition of striosomes is. *Kelly et al., 2017* have found that at E11.5, the time chosen for our tamoxifen induction, neurons expressing D1 dopamine receptors (D1Rs) and those expressing D2 dopamine receptors (D2Rs) are both being born, with a bias toward D1 neurons. Other evidence suggests a predominance of D1R-containing neurons in striosomal mouse models (*Banghart et al., 2015*; *Cui et al., 2014*; *Smith et al., 2016*) or, contrarily, a larger amount of D2R-containing neurons (*Salinas et al., 2016*). It is likely that differential labeling of subtypes of striosomal and matrix neurons occurs in different mouse lines, as has been seen by ourselves (Crittenden and Graybiel, in prep.), and in different regions of the striatum. It is clearly of great interest to determine the neuronal response properties of specific subgroups of striosomal neurons as defined by genetic markers, but we here have chosen to have secure visual identification of striosomal and matrix populations based on the identification of restricted neuropil labeling of striosomes achieved by their birth-dating and confirmed by their correspondence to the classic identification of striosomes in rodents as MOR1-dense zones (*Tajima and Fukuda, 2013*).

## Prospects for future work

Our findings are confined to the analysis of a very simple task, and they clearly are unlikely to have uncovered the range of functions of the striosome and matrix compartments. Yet the experiments do demonstrate the feasibility of definitively identifying striosomes by 2-photon imaging as mice perform tasks, and of examining the activity of striosomal neurons relative to the activity of simultaneously imaged neurons in the nearly matrix. Our findings demonstrate commonality of striosomal and matrix activities during performance of a cued classical conditioning task. The different emphases on reward prediction and reward history that we detected, however, already suggest that striosomal neurons could be more responsive to the immediate contingencies of events than nearby matrix neurons, that they could gain this enhanced sensitivity by virtue of learning-related plasticity, but that they could be less sensitive to immediately prior reward history. These attributes of the striosomes could be related to real-time direction of action plans based on real-time estimates of value. To our best knowledge, this is the first report of simultaneous recording of visually identified striosome and matrix compartments in the striatum, here made possible by the neuropil labeling in pulse-labeled Mash1-CreER mice. Future refinements of such imaging should help to define the functional correlates of the striosome-matrix organization of the striatum.

# Materials and methods

### Key resources table

| Reagent type (species) or resource | Designation | Source or reference | Identifiers | Additional information |
|---|---|---|---|---|
| strain, strain background (mouse,both sexes) | Mash1(Ascl1)-CreER | Jackson Laboratory | Ascl1tm1.1(Cre/ERT2)Jejo/J | Stock no: 12882 |
| strain, strain background (mouse,both sexes) | Ai14 | Jackson Laboratory | B6.Cg-Gt(ROSA)26Sortm14 (CAG-tdTomato)Hze/J | Stock no: 007914 |
| strain, strain background (mouse,both sexes) | C57Bl6/J | Jackson Laboratory | C57BL/6J | Stock no: 000664 |
| genetic reagent | AAV5-hSyn-GCaMP6s-wpre-sv40 | University of Pennsylvania Vector Core) | | |
| antibody | anti-MOR1 | Santa-Cruz | sc-7488 | Polyclonal goat (1:500) |
| antibody | anti-GFP | Abcam | ab13970 | Polyclonal chicken (1:2000) |
| software, algorithm | Matlab | Mathworks | | |
| software, algorithm | Image-J | National Institutes of Health | | |

All experiments were conducted in accordance with the National Institutes of Health guidelines and with the approval of the Committee on Animal Care at the Massachusetts Institute of Technology (MIT).

### Mice

Mash1(Ascl1)-CreER mice (*Kim et al., 2011*) (Ascl1tm1.1(Cre/ERT2)Jejo/J, Jackson Laboratory) were crossed with Ai14-tdTomato Cre-dependent mice (*Madisen et al., 2010*) (B6;129S6-Gt(ROSA)26Sor, Jackson Laboratory) to achieve tdTomato labeling driven by Mash1 and crossed with FVB mice in the MIT colony to improve breeding results. Female Mash1-CreER;Ai14 mice were then crossed with C57BL/6J males to breed the mice that we used for the experiments. Tamoxifen was administered to pregnant dams by oral gavage (100 mg/kg, dissolved in corn oil) to induce Mash1-CreER at embryonic day (E) 11.5, a time point at which predominantly striosomal but almost no matrix neurons are born, in order to label predominantly striosomal neurons in anterior to mid-anteroposterior levels of the caudoputamen. Five mice (4 male and one female) were used for the imaging experiments.

## Surgery

### Virus injections

Adult Mash1(Ascl1)-CreER;Ai14 mice received virus injections during aseptic stereotaxic surgery at 7–10 weeks of age. They were deeply anesthetized with 3% isoflurane, were then head-fixed in a stereotaxic frame, and were maintained on anesthesia with 1–2% isoflurane. Meloxicam (1 mg/kg) was subcutaneously administered, the surgical field was prepared and cleaned with betadine and 70% ethanol, and based on pre-determined coordinates, the skin was incised, the head was leveled to align bregma and lambda, and two holes (ca. 0.5 mm diameter) were drilled in the skull. Two injections of AAV5-hSyn-GCaMP6s-wpre-sv40 (0. 5 μl each, University of Pennsylvania Vector Core) were made, one per skull opening, to favor widespread transfections of striatal neurons at the following coordinates relative to bregma: 1) 0.1 mm anterior, 1.9 mm lateral, 2.7 mm ventral and 2) 0.9 mm anterior, 1.7 mm lateral and 2.5 mm ventral. Injections were made over 10 min, and after a ~10 min delay, the injection needles were slowly retracted. The incision was sutured shut, the mice were kept warm during post-surgical recovery, and they were given wet food and meloxicam (1 mg/kg, subcutaneous) for 3 days to provide analgesia.

### Cannula implantation

We assembled chronic cannula windows by adhering a 2.7 mm glass coverslip to the end of a stainless steel metal tubing (1.6–1.8 mm long, 2.7 mm diameter; Small Parts) using UV curable glue (Norland). Cannula windows were kept in 70% ethanol until used for surgery. At 20–40 days after virus injection, mice were water restricted, and a second surgery was performed under deep isoflurane anesthesia as before to allow insertion of a cannula for imaging (*Dombeck et al., 2010*; *Howe and Dombeck, 2016*; *Lovett-Barron et al., 2014*) and mounting of a headplate to the skull for later head fixation. Bregma and lambda were aligned in the horizontal plane, and the anterior and lateral coordinates for the craniotomy were marked (0.6 mm anterior and 2.1 mm lateral to bregma). The skull was then tilted and rolled by 5° to make the skull surface horizontal at the location of cannula implantation. A 2.7 mm diameter craniotomy was made with a trephine dental drill. The exposed cortical tissue overlying the striatum was aspirated using gentle suction and constant perfusion with cooled, autoclaved 0.01 M phosphate buffered saline (PBS), and part of the underlying white matter was removed. A thin layer of Kwiksil (WPI) was applied, and the chronic cannula was inserted into the cavity. Finally, metabond (Parkell) was used to secure the implant in place and to attach a headplate to the skull. The mice received the same post-surgical care as described above.

## Behavioral training

When mice had recovered from surgery and the optical window had cleared, they were put under water restriction (1–1.5 ml per day) and were habituated to head-fixation for on average 5 days. During head fixation, the mice were held in a polyethylene tube that was suspended by springs. When they showed no clear signs of stress and readily drank water while being head-fixed, behavioral training was begun. Training and imaging was performed 5 days a week. Water was delivered through a tube controlled by a solenoid valve located outside of the imaging setup, and licking at the spout was detected by a conductance-based method (*Slotnick, 2009*). In the behavioral training protocol, two tones (4 or 11 kHz, 1.5 s duration) were played in a random order. The tones predicted reward delivery (5 μl) with, respectively, an 80% or 20% probability. In each trial, there was a 500 ms delay after tone offset before reward delivery. Inter-trial intervals were randomly drawn from a flat distribution between 5.25 and 8.75 s. Training (acquisition phase) was considered to be complete when there was a significant difference in anticipatory licking during the cue period between the two cues (two-sided t-test, $\alpha = 0.05$). Two of the five mice were initially trained on a three-tone version of the task. The training data of these mice have therefore not been included in our analysis. After reaching the acquisition criterion, mice were tested during 4–9 daily session (criterion phase). After completing the criterion phase of the experiment, two mice were given five overtraining sessions (overtraining phase).

## Imaging

Imaging of GCaMP6s and tdTomato fluorescence was performed with a commercial Prairie Ultima IV 2-photon microscopy system equipped with a resonant galvo scanning module and a LUMPlanFL,

40x, 0.8 NA immersion objective lens (Olympus). For fluorescent excitation, we used a titanium-sapphire laser (Mai-Tai eHP, Newport) with dispersion compensation (Deep See, Newport). Emitted green and red fluorescence was split using a dichroic mirror (Semrock) and directed to GaAsP photomultiplier tubes (Hamamatsu). Individual fields of view were imaged using either galvo-resonant or galvo-galvo scanning, with acquisition framerates between 5 and 20 Hz. Laser power at the sample ranged from 11 to 42 mW, depending on GCaMP6s expression levels. For final analysis of the data set, all imaging sessions were resampled at a framerate of 5 Hz.

Fields of view were chosen on the basis of clear labeling of putative striosomes defined by dense tdTomato signal in the neuropil. Within these zones, both tdTomato-positive as well as unlabeled cells were present and were defined as putative striosomal neurons. Because of the 2.4 mm inner diameter of the cannula, we could typically find several striosomes that we could image at different depths. Our sampling strategy was to image as many different neurons as possible. During training, we rotated through the fields of view, but after training and during overtraining, we imaged unique, non-overlapping fields of view.

## Image processing and cell-type identification

Calcium imaging data were acquired using PrairieView acquisition software and were saved into multipage TIF files. Data were analyzed by using custom scripts written in ImageJ (National Institutes of Health) or Matlab (Mathworks). Analysis scripts are available at Github (https://github.com/bloemb/eLife_2017_scripts) (*Bloem, 2017*). Images were first corrected for motion in the X-Y axis by registering all images to a reference frame. We used the pixel-wise mean of all frames in the red channel containing the structural tdTomato signal to make a reference image. All red channel frames were re-aligned to the reference image by the use of 2-dimensional normalized cross-correlation (template matching and slice alignment plugin) (*Tseng et al., 2011*). The green channel frames containing the GCaMP6s signal were then realigned using the same translation coordinates with the 'Translate' function in ImageJ. To verify that calculating translation coordinates on the basis of the tdTomato signal did not provide better registration for striosomal than for matrix neurons, we compared the results obtained by this method with those obtained using a registration method that only uses the GCaMP6s signal. We found that, for both striosomes and matrix, the results for these registration methods were highly correlated (mean correlation coefficient: 0.9971 for striosomes and 0.9978 for matrix). After realignment, ROIs were manually drawn over neuronal cell bodies using standard deviation and mean projections of the movies. With custom Matlab scripts, we drew rings around the cell body ROIs (excluding other ROIs) to estimate the contribution of the background neuropil signal to the observed cellular signal. Fluorescence signal for each neuron was computed by taking the pixel-wise mean of the somatic ROIs and subtracting 0.7x the fluorescence of the surrounding neuropil, as previously described (*Chen et al., 2013*). After this step, the baseline fluorescence for each neuron ($F_0$) was calculated using K-means (KS)-density clustering to find the mode of the fluorescence distribution. The ratio between the change in fluorescence and the baseline was calculated as $\Delta F/F = F_t - F_0 / F_0$. For population analysis of single cell data, we calculated z-scores of the neuronal responses using the mean and the standard deviation of the 1 s baseline period preceding the tone onset.

Individual neurons were identified as striosomal if their cell bodies lay in a region that was densely labeled by tdTomato, or if the cells themselves were tdTomato-positive. Hence, the small minority of tdTomato-positive neurons that appeared in the matrix (*Kelly et al., 2017*) was included in the striosomal population. Altogether 6320 neurons were recorded (2871 during acquisition, 2704 after criterion, and 745 during overtraining). Of these, 1867 were considered striosomal (912 during training, 727 after criterion, and 228 during overtraining). Of these, 294 were labeled with tdTomato, 1828 were located in densely tdTomato-labeled striosomes, and 255 met both criteria. There were 39 tdTomato-labeled cells that were not located in a zone of dense tdTomato neuropil labeling. We excluded these neurons in the multiple analyses resported, but their exclusion never resulted in a different outcome in our analyses.

## Analysis of neuropil activity

To provide a first insight into striosomal and matrix signaling, we integrated the fluorescence signal from within an identified striosome and from a part of the matrix in the same field of view that had a

similar size, background fluorescence and number of neurons. ΔF/F, calculated as $\Delta F/F = F_t - F_0 / F_0$, was normalized by calculating z-scores relative to the signal during the last 1 s of inter-trial intervals to correct for relative differences between sessions. To determine the selectivity of responses to different task events, the area under the Receiver Operating Characteristic curves (AUROC) was calculated. For cue selectivity, we calculated the AUROC by comparing the response during high- and low-probability cues. For the selectivity to rewarded trials, we calculated the AUROC by comparing separately rewarded and unrewarded trials for the two cues.

## Analysis of single-neuron activity

The conditioning task had three epochs — cue, post-reward licking, and post-licking. To identify task-modulated neurons active during these epochs, we aligned the data either to tone onset, to the first lick after reward delivery, or to the end of licking. We compared the fluorescence values over the following time windows to a 1 s baseline preceding each event. For the tone-aligned data, mean fluorescence was calculated over a 2 s time window after tone onset separately for trials with either the high- or low-probability cues. Neurons that were significantly active in either of the cue conditions were considered to be task-modulated. To find neurons modulated during the post-reward licking period, GCaMP6 fluorescence was averaged between the time when the animal first licked to receive the reward and the time that it stopped licking. We also used a 1 s time window after end of licking for identifying task-modulated neurons during this period. In some trials, animals did not stop licking until the start of the next trial. These trials were excluded from the analysis due to the difficulty in assigning licking end-time. For a neuron to be considered as task-modulated, we required that its activity exhibit a significant increase from baseline for any of the three alignments (two-sided Wilcoxon rank-sum test; $\alpha = 0.01$, corrected for multiple comparisons). Neurons exclusively active during only one epoch of the task were considered to be selectively responsive during that period. Most neurons (>80%) were significantly active only during one of the epochs. To compare signals across neurons, we used z-score normalization of the ΔF/F signals with a 1 s period before the cue as a baseline. For analysis of the peak activity of task-modulated neurons, ΔF/F signals were normalized to the maximum of the session-averaged activity for any particular alignment in order to compare peak activity times during the time interval of interest. For determining the temporal specificity of responses during the post-reward licking period (rewarded trials with high-probability cue), we generated shuffled data for each neuron by substituting the response in a given trial with response in the same trial from a randomly selected task-modulated neuron recorded simultaneously. Only sessions in which at least ten task-modulated neurons were simultaneously recorded were included in this analysis. We computed a reliability index defined as the average response correlation of all pairwise combinations of trials (*Rikhye and Sur, 2015*). In addition, we quantified the standard deviation of peak response times across trials. For these measurements, we repeated the shuffle 20 times for each neuron and calculated the mean value of the outcome of the 20 shuffled analyses as the representative metric. Significance was then computed by comparing the observed and shuffled distribution of values using a Wilcoxon rank-sum test. We also computed a ridge-to-background ratio (*Harvey et al., 2012*), which quantifies the relative magnitude of response close to the peak time relative to all other time points during the post-reward period. The ridge was defined as the mean ΔF/F value (normalized to the max response) taken over five time points (i.e., 1 s due to the 5 Hz frame acquisition rate of our recordings) surrounding the peak time for each neuron's session-averaged response, and the background value was the mean ΔF/F over all other time points.

To determine whether reward outcome in the previous trial modulated licking behavior during the current trial, we first compared anticipatory licking in trials that were followed by either rewarded or unrewarded trials. We included all current trials, regardless of the cue or the outcome status. To examine the effect of outcome history on licking after reward delivery, we analyzed only currently rewarded trials, again ignoring the identity of the cue presented. To determine whether neural responses were modulated by previous outcome history, we computed a history modulation index (HMI) using the following formula:

$$\text{HMI} = \frac{\text{Previous trial rewarded} - \text{Previous trial unrewarded}}{\text{Previous trial rewarded} + \text{Previous trial unrewarded}}$$

The HMI was computed from z-score values normalized by the following method. First, we took

all currently rewarded trials and averaged the z-scores of ΔF/F values over a 2 s window starting 1 s after reward delivery. We chose this time window because we found that most of the task-modulated neurons were active during this period. These values were then scaled by the range of the observed responses, so that normalized values ranged from 0 to 1. Trials were then separated based on different outcome histories.

### Linear regression analysis

To quantify the relationship between behavioral performance and neuronal activation, we used linear regression. For every session, we calculated the baseline licking and ΔF/F activation in the 1 s period preceding the cue onset and calculated the mean standard deviation of the baseline across trials, which we then used to calculate z-scores of the tone-evoked licking and ΔF/F activation for every trial. We then averaged the normalized tone-evoked licking and ΔF/F response across trials for both cue types for every session. Next, we performed linear regression analyses to identify a possible relationship between tone-evoked ΔF/F activation and tone-evoked licking. We performed this regression for both high- and low-probability tones and for the difference in the licking and ΔF/F responses between them. As a first step, we created separate models for striosomes and matrix in order to calculate the regression coefficients and significance for these populations separately. In order to compare striosomes and matrix more directly, we made a combined model and then quantified the residuals for striosomes and matrix. The differences in residuals were compared using a paired t-test.

### Statistical analysis

We used Wilcoxon sign-rank tests to detect significant modulation of single neurons in different task epoch. ANOVA was used to evaluate interactions between multiple factors. For percentages, Fisher's exact test was used to compare groups, and confidence intervals were calculated using binomial tests.

### Histology

After the experiments, mice were transcardially perfused with 0.9% saline solution followed by 4% paraformaldehyde in 0.1 M NaKPO$_4$ buffer (PFA). The brains were removed, stored overnight in PFA solution at 4°C and transferred to glycerol solution (25% glycerol in tris buffered saline) until being frozen in dry ice and cut in transverse sections at 30 μm on a sliding microtome (American Optical Corporation). For staining, sections were first rinsed 3 × 5 min in PBS-Tx (0.01 M PBS + 0.2% Triton X-100), then were incubated in blocking buffer (Perkin Elmer TSA Kit) for 20 min followed by incubation with primary antibodies for GFP (Polyclonal, chicken, Abcam ab13970, 1:2000) and MOR1 (Polyclonal, goat, Santa Cruz sc-7488, 1:500). After two nights of incubation at 4°C, the sections were rinsed in PBS-Tx (3 × 5 min), incubated in secondary antibodies Alexa Fluor 488 (donkey anti-chicken, Invitrogen, 1:300) and Alexa Fluor 647 (donkey anti-goat, Invitrogen, 1:300) for 2 hr at room temperature, rinsed in 0.1 M PB (3 × 5 min), mounted and covered with a coverslip with ProLong Gold mounting medium with DAPI (Thermo Fisher Scientific).

To quantify the overlap between striosomes as detected by tdTomato and MOR1 staining, we stained sections from five mice and recorded images of 2 brain sections per mouse. We manually outlined striosomes for every marker twice and calculated the percentage of pixels that were marked as striosomes and matrix. In addition, we compared the repeated outlines of the striosomes that were made using the same marker, allowing us to get a measure of test-retest error rates when outlining striosomes on the basis of tdTomato or MOR1.

## Acknowledgements

We thank Dr. Mark Howe and Dr. Dan Dombeck for invaluable advice on 2-photon imaging of the striatum, Dr. Leif Gibb and Jannifer Lee for initiating the breeding program, Dr. Josh Huang and Dr. Sean Kelly for their advice in this process, Cody Carter for critical help with the breeding of the mice, Dr. Yasuo Kubota for help preparing the manuscript, and Erik Nelson for his work on the histology.

# Additional information

## Funding

| Funder | Grant reference number | Author |
|---|---|---|
| Simons Foundation | 306140 | Ann M Graybiel |
| National Institute of Mental Health | R01 MH060379 | Ann M Graybiel |
| Saks Kavanaugh Foundation | | Ann M Graybiel |
| Bachmann-Strauss Dystonia and Parkinson Foundation | | Ann M Graybiel |
| Netherlands Organization for Scientific Research - Rubicon | | Bernard Bloem |
| National Institute of Neurological Disorders and Stroke | U01 NS090473 | Mriganka Sur |
| National Eye Institute | R01 EY007023 | Mriganka Sur |
| National Science Foundation | EF1451125 | Mriganka Sur |
| Simons Foundation Autism Research Initiative | | Mriganka Sur |
| National Eye Institute | F32 EY024857 | Rafiq Huda |
| National Institute of Mental Health | K99 MH112855 | Rafiq Huda |
| Nancy Lurie Marks Family Foundation | | Ann M Graybiel |
| William N. & Bernice E. Bumpus foundation | RRDA Pilot: 2013.1 | Ann M Graybiel |
| William N. & Bernice E. Bumpus foundation | | Bernard Bloem |

The funders had no role in study design, data collection and interpretation, or the decision to submit the work for publication.

## Author contributions

Bernard Bloem, Rafiq Huda, Conceptualization, Software, Formal analysis, Funding acquisition, Investigation, Visualization, Methodology, Writing—original draft, Writing—review and editing; Mriganka Sur, Resources, Supervision, Funding acquisition, Writing—review and editing; Ann M Graybiel, Conceptualization, Resources, Supervision, Funding acquisition, Methodology, Writing—original draft, Writing—review and editing

## Author ORCIDs

Bernard Bloem http://orcid.org/0000-0002-0930-580X
Ann M Graybiel http://orcid.org/0000-0002-4326-7720

## Ethics

Animal experimentation: All experiments were conducted in accordance with the National Institute of Health guidelines and with the approval of the Committee on Animal Care at the Massachusetts Institute of Technology (protocol #: 1114-122-17).

## Decision letter and Author response

Decision letter https://doi.org/10.7554/eLife.32353.022
Author response https://doi.org/10.7554/eLife.32353.023

## Additional files

**Supplementary files**

• Transparent reporting form
DOI: https://doi.org/10.7554/eLife.32353.020

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
