## [Decision Letter]

Thank you for submitting your article "Two-Photon Imaging of Striatum Demonstrates Distinct Functions for Striosomes and Matrix in Reinforcement Learning" for consideration by *eLife*. Your article has been favorably evaluated by Eve Marder (Senior Editor) and three reviewers, one of whom, Geoffrey Schoenbaum (Reviewer #1), is a member of our Board of Reviewing Editors.

The reviewers have discussed the reviews with one another and the Reviewing Editor has drafted this decision to help you prepare a revised submission.

Summary:

In this paper, the authors deploy a new approach for separating neural signals from patch and matrix neurons recorded in awake, behaving head fixed mice. Using 2-photon Ca^++^ imaging to separately image these populations in mice performing an auditory discrimination for water reward, they identify neural correlates of cue, response and reward in both populations. Striosomal activity is somewhat stronger and discriminates reward better but overall the simple task identifies very similar neural correlates of the behaviors.

Essential revisions:

All three reviewers agreed that the paper is a technical marvel and presents a potentially transformative tool for the study of the different circuits that run through these two compartments of striatum. This has long been a question that has interested researchers, and the authors seem to have a strong approach for dissociating them in awake behaving animals. This is extraordinary. However all three reviewers felt that additional detail regarding the specificity and sensitivity of the dissociation was needed, as well as some hand-holding perhaps to explain it better, since it is such a novel and important part of the study. Specific requests are given by reviewer 1 and 2 in their initial comments.

The reviewers also generally agreed that the responses of patch and matrix cells were remarkably similar. The authors understandably emphasized the differences, however they seemed relatively slight against a background of very similar activity. Obviously this is somewhat due to the very simple structure of the task used, but it was felt the treatment of this should be more balanced. Indeed isn't it perhaps more interesting that there are so few differences, given all the strong proposals about these two compartments? Further there was some concern that the differences observed could be related to signal differences or normalization approaches (reviewers 1and 2, second comment and reviewer 3, first comment). This should be addressed and the data presented in a more balanced fashion (Abstract, title, Discussion).

Related but separate from this, it was also felt that the authors should also be more clear as to how they think the differences or lack thereof in their data affect any theoretical accounts. To quote one reviewer from the discussion: "It would be a great opportunity for the Graybiel lab to add some insight to the literature on their current thoughts on the function of these compartments" given their data – whether there turn out to be differences – or maybe even more interestingly if the differences are deemed not the main story.

Finally all three reviewers had some difficulty following what was done. This is a general problem throughout, but is particularly acute for understanding how many mice contributed to the recordings of the different cell populations in the different periods of training. This is indicated in strongly by reviewer 2, third point, and reviewer 3, fourth point. This should be clarified.

*Reviewer #1:*

In this study the authors used 2-photon calcium imaging combined with transgenically-restricted birthdate-labeling of striosomal neurons to gather data on the differential activity of striosome and matrix neurons in the DS of head-fixed mice performing a simple auditory discrimination task for water. This is a pressing question given the differential connectivity of these two compartments and the speculative theories regarding their differential functions. As the authors note, there is little or no data from neural correlates substantiating any of these exciting ideas. Calcium imaging combined with genetic techniques to discriminate the two regions offers the possibility of addressing this potentially important question. Using this approach, the authors seem to have the ability to distinguish patch and matrix neurons in mice learning the discrimination task. They find both populations show task-related firing. Activity is related to the cues, rewards and post-reward period. It is somewhat stronger and discriminates reward better in the striosomes, but neurons in both compartments show generally similar patterns of activity.

Overall the paper is a technical tour de force. Combining calcium imaging with the fate-mapping to segregate these neural compartments is brilliant and offers a potential tool that can be used to test the various theories that have been advanced for how they interact to support striatal function. I think it could be improved if the authors would provide a clearer explanation for how this works for the uninitiated and a more detailed accounting of the specificity of this method (cell counts of numbers of labeled neurons in the patches, out of the patches, versus unlabeled). However it looks very promising.

In this context, the authors chose to apply a very basic task. As they note, this is just a first step, but perhaps as a result, the differences identified are relatively slight. In fact, to me, the patterns are remarkably similar in the two cell types, especially when one considers that the striosomes seem overall more active. A higher overall level of activity would give rise to many of the other statistical differences it seems to me, such as somewhat higher percentages of statistically engaged neurons in a particular epoch or better discrimination of reward vs. non-reward. These differences may be important to a downstream observer, but they are clearly modest and entirely quantitative, rather than qualitative. It seems to me that the truly transformative theoretical accounts mentioned would predict serious qualitative differences in the right setting.

Of course the authors note that they applied a simple task intentionally. The behavioral approach and analysis (at least as described) was not intended to directly target the predictions of any of these accounts, but merely to test in a simple setting whether there were any differences. But as a result, it seems to me that the data do not really challenge or force a modification of any of these proposals. Or at least it is unclear to me whether the authors believe they do – in other words, are the authors prepared to say that their findings cast doubt on or favor any of these proposals that the patches and matrix neurons do fundamentally different jobs? I did not get the impression that the data do this, either from reading them, or from the author's Discussion.

So overall my impression is that this is a remarkable paper in terms of the tools it applies and technical approaches, but the heavy lifting (as the authors note) is left to future studies. My feeling is that this is more than sufficient because of the importance of distinguishing these populations and the novelty of this approach.

*Reviewer #2:*

This paper addresses an interesting and understudied aspect of striatal complexity, namely how the striosome and matrix components of the striatum function in reinforcement learning. I found the results interesting but largely descriptive. There were no attempts to manipulate function of these populations and thereby test their necessity in these behaviors. As such, I was left unclear on whether these populations have a distinct function in reinforcement learning beyond the correlative differences the authors' observed in their recordings. I was also concerned about the number of animals used, particularly in the over-training dataset where n=2. Additional methodological issues may also impact their results and could be clarified. Specifically:

1) There was little quantitative information on how well their manipulation targeted striosomes. I would like to see a zoomed out image of the striatum, as well as quantification of MOR overlap that they mentioned. I was also concerned that <20% of the neurons labeled as striosomes with their strategy actually expressed tdTmt. What does this say about their expression strategy? Were quantitative methods employed to split non-labeled neurons into striosome/matrix?

2) Did the recording quality (ΔF/F, mean fluorescence, overall variance) differ between the striosome and matrix identified neurons? Did the presence of tdTmt in labeled cells impact the quality of the GCaMP recordings in those cells?

3) Some description of consistency across mice is warranted throughout the paper. Were similar proportions of striosome/matrix neurons recorded in all mice? Was the size and quality of neuronal responses consistent across animals? If not, I worry that results may reflect differences between animals, rather than differences between cell types. This is especially worrisome in the overtraining data, where n=2 mice.

4) Although licking behavior itself changes with training, this was not discussed in the context of their results. Particularly in the over-training experiments, are the neuronal responses to the tone and licking co-varying with increases in licking? Or are they independent?

*Reviewer #3:*

The authors address a long-standing question about the different functions of striosomal vs. matrix neurons in striatum. Using a new mouse line the authors have recorded striosomal neurons for the first time, a major achievement! Based on extensive anatomical evidence, largely by the senior author, striosomes have been proposed to serve an evaluation function during reinforcement learning, therefore I would have expected rather distinct responses. The authors did find some differences between striosome vs. matrix neurons, but the major conclusion for me was that they appeared rather similar. I think this should be communicated in the manuscript. There are also some technical issues that need to be addressed to support the conclusions about differences.

1) One of their central findings is that striosomal SPNs provide more selective reward responses compared to their matrix counterparts. They show that normalized striosomal responses to reward-predictive cues and to reward are stronger. One potential issue lies in how this normalization is conducted. The authors use z-score normalization to compare across neurons and conditions. One possibility is the enhanced responses are due to increases in activity in response to task events; alternatively they are due to decreased baseline variability. Given that the authors use the striosomal-specific red fluorescence channel to align their image, it's possible that measured neuronal (not neuropil) activity within the matrix is subject to increased noise from imperfect alignment. This issue with baseline variability is visible in the example neurons shown in 5D. The proportion of striosomal vs. matrix task-modulated neurons might also be affected by differences in the noise floor.

2) It would be informative to quantify the variability in timing of responses across trials for both striosomal and matrix populations.

3) The finding that striatal SPNs "tiled the temporal space of the task" needs to be backed up by appropriate controls. Is the apparent tiling of temporal space in the 2d plots in 4G and 5F due to variability inherent in measuring this timing given limited numbers of trials? The authors could address this by quantifying the peak time and spread (standard deviation or other metric) of each neuron. Then they can compare these values to a synthetic population of neurons by shuffling neuron labels for each trial.

4) The division and analysis of neuronal responses across acquisition, criterion and overtraining phases is useful and informative, but also somewhat arbitrary. It would be more informative to quantify how responses evolve on a more continuous basis relative to conditioned behavior. Relatedly, they could explicitly examine stimulus-evoked activity by performing a linear regression to model how activity is explained by licking behavior. The expectation is that the residuals from this fit would be enhanced for striosomal SPNs for cue and reward relative to matrix SPNs.

The demonstration that matrix neurons are modulated more strongly by reward history is nice and begins to address the issue with increased response magnitude vs. baseline variability I have raised above.

---

## [Author Response]

Essential revisions:All three reviewers agreed that the paper is a technical marvel and presents a potentially transformative tool for the study of the different circuits that run through these two compartments of striatum. This has long been a question that has interested researchers, and the authors seem to have a strong approach for dissociating them in awake behaving animals. This is extraordinary. However all three reviewers felt that additional detail regarding the specificity and sensitivity of the dissociation was needed, as well as some hand-holding perhaps to explain it better, since it is such a novel and important part of the study. Specific requests are given by reviewer 1 and 2 in their initial comments.The reviewers also generally agreed that the responses of patch and matrix cells were remarkably similar. The authors understandably emphasized the differences, however they seemed relatively slight against a background of very similar activity. Obviously this is somewhat due to the very simple structure of the task used, but it was felt the treatment of this should be more balanced. Indeed isn't it perhaps more interesting that there are so few differences, given all the strong proposals about these two compartments? Further there was some concern that the differences observed could be related to signal differences or normalization approaches (reviewers 1and 2, second comment and reviewer 3, first comment). This should be addressed and the data presented in a more balanced fashion (Abstract, title, Discussion).

We have greatly benefitted from these comments. We agree that we should in our original submission have commented more fully on how similar the responses of the sampled striosome and matrix regions were found to be. In fact, we now have featured this similarity alongside the quantitative differences that we observed. We now note in the manuscript that the very similarity adds to the need for further work to identify what key functional properties do distinguish them.

Related but separate from this, it was also felt that the authors should also be more clear as to how they think the differences or lack thereof in their data affect any theoretical accounts. To quote one reviewer from the discussion: "It would be a great opportunity for the Graybiel lab to add some insight to the literature on their current thoughts on the function of these compartments" given their data – whether there turn out to be differences – or maybe even more interestingly if the differences are deemed not the main story.

We appreciate this request. We have added to the discussion of what these compartments might do, but we are limited in how much such discussion we can add because we realize – as the reviewers did, but we inadvertently failed to write adequately – that we have not yet delineated the functions posited for these compartments. We are adding that greater responsivity to predictors of reward in the response profiles of the sampled striosomal neurons would be in line with striosomes acting as critic in an actor-critic architecture, but this finding does not nail this as their exclusive function. This heightened sensitively would also, for example, be in accord with the responsibility function idea that our lab put forward, along with other ideas based on limbic associations of the striosomes. We introduce this manuscript to determine very basic features of the two compartments with what we believe to be the first imaging study, combining 2-photon imaging with birthdate labeling enabling neuropil plus cell body labeling of striosomal cells so that striosomal and matrix neurons can be simultaneously imaged. We definitely hope to go on with a deep exploration of these issues.

Finally all three reviewers had some difficulty following what was done. This is a general problem throughout, but is particularly acute for understanding how many mice contributed to the recordings of the different cell populations in the different periods of training. This is indicated in strongly by reviewer 2, third point, and reviewer 3, fourth point. This should be clarified.

We deeply apologize for having been unclear. In the revision that we have prepared, we have made every effort to incorporate missing experimental details not only in the text but also in a new set of tables with information about the recorded neurons (Table 2) and sessions per mouse (Table 4). We address this issue further in response to relevant reviewers’ questions below.

We have also performed a series of additional analyses described in the revised manuscript. We have added new panels in Figure 6 and Figure 7 that address the question 4 of reviewer 2 and have added supplementary material addressing the question 1 of reviewer 2 (Figure 1—figure supplement 1; Table 1), the questions from reviewers 1 (question 3) and 2 (question 2) (Table 2), and the questions 2, 3 and 4 of reviewer 3 (Figure 4, Figure 5 and Figure 7—figure supplement 1; Table 5).

Reviewer #1:[…] Overall the paper is a technical tour de force. Combining calcium imaging with the fate-mapping to segregate these neural compartments is brilliant and offers a potential tool that can be used to test the various theories that have been advanced for how they interact to support striatal function. I think it could be improved if the authors would provide a clearer explanation for how this works for the uninitiated and a more detailed accounting of the specificity of this method (cell counts of numbers of labeled neurons in the patches, out of the patches, versus unlabeled). However it looks very promising.

We thank the reviewer very much for these positive comments. We have now added more details about the birth-dating method and apologize for not having been clearer in our original submission. The key to this protocol is that by pulse-tagging the neurons in the mouse line that we used (by administering tamoxifen to pregnant dams), we could not only achieve strong tdTomato labeling of a group of neurons born within the time-window of neurogenesis of striosomal cells, but also achieve labeling of the local processes of these cells, which are known to remain mainly within the bounds of striosomes. This meant that even though the tdTomato marker labeled only a small proportion of the cells that would eventually differentiate into striosomal cells at maturity, they had sufficient local processes to define the borders of striosomes. With this method, we found relatively few tdTomato-labeled neurons outside of striosomes, namely, 19/1996 total matrix cells detected as compared to 111/708 total cells detected in the striosomal sample. We carefully confirmed the match to striosomes by immunohistochemistry (for which we have added further illustrations in Figure 1—figure supplement 1; Table 1). We have added a table summarizing the number of labeled neurons inside and outside of striosomes for the total population and for the individual mice (Table 2). We have also added description of the birth-dating method, both in the Results and in the Materials and methods sections.

In this context, the authors chose to apply a very basic task. As they note, this is just a first step, but perhaps as a result, the differences identified are relatively slight. In fact, to me, the patterns are remarkably similar in the two cell types, especially when one considers that the striosomes seem overall more active. A higher overall level of activity would give rise to many of the other statistical differences it seems to me, such as somewhat higher percentages of statistically engaged neurons in a particular epoch or better discrimination of reward vs. non-reward. These differences may be important to a downstream observer, but they are clearly modest and entirely quantitative, rather than qualitative. It seems to me that the truly transformative theoretical accounts mentioned would predict serious qualitative differences in the right setting.

The reviewer has helped us greatly by these comments. We did exactly as he/she said: we purposely used a simple classical conditioning task in this first report on the detection of striosomal and matrix neurons by direct imaging, as we intended to determine whether the method could reliably detect basic features of striatal function as they have been reported in many previous papers relating the activity of striatal SPNs to aspects of reinforcement-directed behavior. We now, in response to the reviewer’s comment, have addressed the question of how differences in baseline activity in striosomal and matrix neurons affect our results. We think that there are two main arguments why general differences in activity do not account for the observed differences in both compartments. First, we find that in ΔF/F normalized data, there are no differences between striosomal and matrix neurons in the mean, the standard deviation, and the peak signal during baseline periods, suggesting similar signal-to-noise ratios in both populations. We have added a table with these measurements (Table 3). Secondly, we find that striosomes and matrix have similar percentages of reward-modulated neurons and that the average responses in both populations are similar. If striosomal neurons were to be simply more active in general, then we would expect striosomes to have more task-modulated neurons during all epochs of the task, and we would expect larger overall responses. But we do not find these patterns. For these reasons, we do not think that differences between the two compartments in general activity levels can account for our findings that striosomes show preferential cue encoding.

As to the theoretical implications of this work, we have tried to write more about these in the Discussion section. We again hope to emphasize that this manuscript is aimed at reporting for what we believe is the first direct simultaneous detection of the activity of striosomes and matrix.

Of course the authors note that they applied a simple task intentionally. The behavioral approach and analysis (at least as described) was not intended to directly target the predictions of any of these accounts, but merely to test in a simple setting whether there were any differences. But as a result, it seems to me that the data do not really challenge or force a modification of any of these proposals. Or at least it is unclear to me whether the authors believe they do – in other words, are the authors prepared to say that their findings cast doubt on or favor any of these proposals that the patches and matrix neurons do fundamentally different jobs? I did not get the impression that the data do this, either from reading them, or from the author's Discussion.

Once again, we agree with the reviewer, and we very much appreciate this comment. We have chosen to do a simple reinforcement task so that we could observe striosomal and matrix activity in a basic task that captures important aspects of striatal function but that is easy to interpret. We believe that a stronger encoding of reward-predicting cues is an important finding that is relevant to many theories of striatal function, but we also acknowledge that we have not yet tested one of the proposals that striosomes and matrix do fundamentally different jobs. We are therefore trying to be careful about drawing conclusions too strongly about the functions of the compartments and trying to stay to the basic observation as close as possible. At the same time, we have found clear quantitative differences in the activities of the two compartments and have tried to report these carefully. We have tried to connect our findings more to the existing literature and hypotheses in the Discussion. In future work, we aim to address the existing hypotheses about striosomal function more directly.

So overall my impression is that this is a remarkable paper in terms of the tools it applies and technical approaches, but the heavy lifting (as the authors note) is left to future studies. My feeling is that this is more than sufficient because of the importance of distinguishing these populations and the novelty of this approach.

This reviewer has been extremely helpful to usin leading us to state directly that this is not a manuscript to “solve” the riddle of striosome-matrix functions; in fact, this manuscript augments this riddle by indicating that there are many commonalities between the striosomes and matrix in basic classical reward-based learning.

Reviewer #2:This paper addresses an interesting and understudied aspect of striatal complexity, namely how the striosome and matrix components of the striatum function in reinforcement learning. I found the results interesting but largely descriptive. There were no attempts to manipulate function of these populations and thereby test their necessity in these behaviors. As such, I was left unclear on whether these populations have a distinct function in reinforcement learning beyond the correlative differences the authors' observed in their recordings. I was also concerned about the number of animals used, particularly in the over-training dataset where n=2. Additional methodological issues may also impact their results and could be clarified. Specifically:1) There was little quantitative information on how well their manipulation targeted striosomes. I would like to see a zoomed out image of the striatum, as well as quantification of MOR overlap that they mentioned. I was also concerned that <20% of the neurons labeled as striosomes with their strategy actually expressed tdTmt. What does this say about their expression strategy? Were quantitative methods employed to split non-labeled neurons into striosome/matrix?

The reviewer raises a critical issue for which we thank him/her. In response to this comment, we have now added a figure supplement containing zoomed out images of the striatum (Figure 1—figure supplement 1), as well as a quantification of the overlap of striosomes as marked by tdTomato and MOR1 staining (Table 1). We find that 2% of the pixels are scored as striosomes on the basis of tdTomato but not MOR1 and that 3.7% of the pixels are scored as striosomes on the basis of MOR1 but not tdTomato. To control for inevitable errors in outlining these structures, we also quantified the test-retest error for outlining MOR1 and tdTomato structures, which we found to be 2.3 and 2.4%, respectively. This shows that using tdTomato will mainly cause us to have false negatives, or missed striosomes, in our analysis.

To answer the reviewer’s question about the small percentage of labeled neurons, the pulse labeling was applied to the mid-interval of neurogenesis of striosomal cells. They are born over a period of ~3 days during embryonic development. We could have given multiple tamoxifen injections on several days to label more neurons, but we chose not to do so because of the adverse effects that this protocol has on the animals, and because this could have resulted in higher false-positive rates, as toward the end of the striosomal neurogenic window, matrix neurons are also beginning to be born. We have experience with the type of anatomical labeling that comes from this sort of pulse tagging, and here again found that the neuropil of the pulse-labeled striosomal neurons largely is confined to striosomes as confirmed with immunostaining. We could obtain sufficient labeling of these local processes of the striosomal neurons to permit detection of striosomal borders, as the illustrations show. We again would like to thank the reviewer for asking for further illustrations and detailed explanation. Our identification of neurons as striosomal critically depended on this neuropil labeling. The delineation of the striosomes was done manually but blind to the neuronal responses.

2) Did the recording quality (ΔF/F, mean fluorescence, overall variance) differ between the striosome and matrix identified neurons? Did the presence of tdTmt in labeled cells impact the quality of the GCaMP recordings in those cells?

We thank the reviewer for addressing this issue. Our original manuscript should have had more information about this, and we apologize. We have now added a paragraph (subsection “Imaging of striosomes”, last paragraph) describing our efforts to control for these issues as well as a table summarizing parameters that are useful for answering this question to the manuscript (Table 3). We find a slightly lower level of expression of GCaMP6 in striosomes. It is not clear what the reason for this is, but we know from the literature as well as from our own experience that many AAVs tend to have lower fluorescent marker signals in striosomes. To overcome this limitation, we normalized the fluorescence signals to their baseline using ΔF/F normalization, as is common in calcium imaging. We compared striosomal and matrix signals by quantifying the mean, the standard deviation and the peak of the signals during the baseline period. We have not found any significant differences with respect to these parameters when we compared matrix neurons to tdTomato-labeled striosomal neurons or neurons within the striosomal neuropil.

3) Some description of consistency across mice is warranted throughout the paper. Were similar proportions of striosome/matrix neurons recorded in all mice? Was the size and quality of neuronal responses consistent across animals? If not, I worry that results may reflect differences between animals, rather than differences between cell types. This is especially worrisome in the overtraining data, where n=2 mice.

We would like to apologize for not including these data in the manuscript. We have now added a table (Table 2) with the number of neurons recorded in every mouse. We tried to get roughly the same numbers of neurons for the different mice, but this was not always possible, because some mice yielded more fields of view in which we could clearly define striosomes than did others.

To address the issue of consistency of recording quality among mice and the low number of mice that we have in the training and overtraining analysis, we have added another table (Table 4). We have included the number of sessions to which the different mice contribute as well as measurements of baseline signals. We find that the size and the quality of the recordings are not the same for all mice, which is likely related to the imaging quality, imaging depth and other differences among mice. However, the mice that were included in Figure 6 and Figure 7 are representative of all mice that were studied.

We thought that the fairest comparison across mice would be to compare the proportions of task-modulated neurons that we found in the different mice. We found that in all mice, the percentage of striosomal and matrix neurons that were task-responsive were roughly similar. In addition, we found that in all mice, there were more cue-modulated neurons in striosomes than in the matrix. Finally, to respond to the question of whether differences among mice, rather than differences between cell types, can account for our results, we would like to note that we found similar results when we analyzed the overall ΔF/F from striosomal/matrix regions. For this analysis, we had matched striosomal and matrix recordings made at the same time, so that differences among mice cannot account for our findings.

4) Although licking behavior itself changes with training, this was not discussed in the context of their results. Particularly in the over-training experiments, are the neuronal responses to the tone and licking co-varying with increases in licking? Or are they independent?

We thank the reviewer foraddressing this interesting issue. We have added plots with the licking data to Figure 6 and to Figure 7 (new panel B) so that readers can directly compare licking behavior with the neuronal data.

Concerning the second part of the comment, interestingly, we find that the neuronal responses and licking during overtraining are dissociable. After prolonged training, the mice initially respond to both cues by licking, but for the low-probability cue, this lick rate goes down during the tone and delay period, whereas with the high-probability cue the mice keep on licking at the same rate. The neuronal data, on the other hand, seem to show the opposite pattern. There is a higher activation after the high-probability cue, but the ΔF/F signal goes down rapidly. For the low-probability cue, even though the response is smaller, the ΔF/F signal stays more or less the same during the cue and the reward-delay period.

Reviewer #3:The authors address a long-standing question about the different functions of striosomal vs. matrix neurons in striatum. Using a new mouse line the authors have recorded striosomal neurons for the first time, a major achievement! Based on extensive anatomical evidence, largely by the senior author, striosomes have been proposed to serve an evaluation function during reinforcement learning, therefore I would have expected rather distinct responses. The authors did find some differences between striosome vs. matrix neurons, but the major conclusion for me was that they appeared rather similar. I think this should be communicated in the manuscript. There are also some technical issues that need to be addressed to support the conclusions about differences.

We are grateful indeed for the remarks of the reviewer. As we have noted in our responses above, we realize that we did not emphasize enough the similarities in responses of the striosomal and matrix neurons that we sampled in these experiments. We have now remedied this problem by many comments throughout the manuscript, and we have also edited the Abstract and title to this end.

1) One of their central findings is that striosomal SPNs provide more selective reward responses compared to their matrix counterparts. They show that normalized striosomal responses to reward-predictive cues and to reward are stronger. One potential issue lies in how this normalization is conducted. The authors use z-score normalization to compare across neurons and conditions. One possibility is the enhanced responses are due to increases in activity in response to task events; alternatively they are due to decreased baseline variability. Given that the authors use the striosomal-specific red fluorescence channel to align their image, it's possible that measured neuronal (not neuropil) activity within the matrix is subject to increased noise from imperfect alignment. This issue with baseline variability is visible in the example neurons shown in 5D. The proportion of striosomal vs. matrix task-modulated neurons might also be affected by differences in the noise floor.

We would like to thank the reviewer for addressing these important issues. We agree that the method of normalization is critical for this study. We have carefully considered this issue and have added a table with information about baseline fluorescence in striosomal and matrix neurons (Table 3). We find that there are no differences between striosomes and matrix regarding the means, standard deviations or maxima of the baseline ΔF/F values, indicating that the observed differences between striosomes and matrix cannot be accounted for by differences in the mean and standard deviation that were used to calculate the z-scores.

In response to the reviewer’s question about alignment, we are also grateful to have the chance to add to our revision. To test how the realignment using the two different channels affects the signals in the striosomes and matrix, we have realigned the GCaMP recordings on the basis of the translation coordinates that were calculated using each channel and have then correlated the fluorescent traces from individual neurons using both alignment methods. We have found very high correlations between the measurements, and these were similar for striosomes and matrix (r = 0.9971 for striosomes and 0.9978 for matrix). These results are now included in the Materials and methods section of the manuscript.

2) It would be informative to quantify the variability in timing of responses across trials for both striosomal and matrix populations.

We thank the reviewer for this suggestion. We have now performed additional analysis to address this issue. We computed a reliability index, which captures the trial-to-trial variability in responses during the three epochs of the task for striosomal and matrix neurons. Reliability was measured by taking the average correlation of all pairwise combinations of rewarded high-probability cue trials. This analysis demonstrated no differences in the mean response variability between striosomal and matrix neurons.

3) The finding that striatal SPNs "tiled the temporal space of the task" needs to be backed up by appropriate controls. Is the apparent tiling of temporal space in the 2d plots in 4G and 5F due to variability inherent in measuring this timing given limited numbers of trials? The authors could address this by quantifying the peak time and spread (standard deviation or other metric) of each neuron. Then they can compare these values to a synthetic population of neurons by shuffling neuron labels for each trial.

We thank the reviewer for raising this important issue. We followed the reviewer’s advice and computed the standard deviation of peak response times for neurons that were active during the post-reward epoch. We only analyzed sessions in which at least ten post-reward active neurons were recorded simultaneously. For each active neuron, we constructed a ‘shuffled’ data set by shuffling the neuron labels (other active neurons within the same population) for each trial 20 times and determined the standard deviation of peak response time (across trials) for each of these shuffles. These 20 values were averaged to estimate the standard deviation for each shuffled data set. Comparing standard deviation values from observed and shuffled data showed more variability in the shuffled data.

As a related analysis, we used the same shuffling procedure and calculated the reliability of responses for observed and shuffled data. This analysis showed that observed data had more reliable responses than the shuffled data. Sorting data by peak time is expected to artificially create a tiling effect. Hence, we compared sorted observed and shuffled responses by quantifying the ridge-to-background ratio, which measures the relative magnitude of responses close to the peak relative to other time points during the trial. Observed data had a higher ratio than the shuffled data. Together, these control analyses suggest that there is structure in the timing of single-neuron responses during the task.

4) The division and analysis of neuronal responses across acquisition, criterion and overtraining phases is useful and informative, but also somewhat arbitrary. It would be more informative to quantify how responses evolve on a more continuous basis relative to conditioned behavior. Relatedly, they could explicitly examine stimulus-evoked activity by performing a linear regression to model how activity is explained by licking behavior. The expectation is that the residuals from this fit would be enhanced for striosomal SPNs for cue and reward relative to matrix SPNs.

We thank the reviewer for suggesting this wonderful idea. We have followed this recommendation and have added the results of this analysis to Figure 7 in the form of a supplement (Figure 7—figure supplement 1) and a table (Table 5), and to the text (subsection “During overtraining, tone-related responses of striosomal neurons intensify and become increasingly selective for high-probability tones”, last paragraph). To summarize, we have tried to predict neuropil ΔF/F responses on the basis of tone-evoked licking. We did this separately for both tones and also for the difference in response between the two tones. First, we made separate models for striosomal and matrix responses and found that in both compartments, tone-evoked licking is a significant predictor of the neuropil response in the case of the high-probability tone but not the low-probability tone. In addition, the difference in licking between the two tones was a significant predictor of the difference in the striosomal response between the cues, but not in the matrix. Secondly, we made one model for combined striosomal and matrix responses and then quantified the residuals of both compartments. Here we found that the residuals of the striosomes are significantly larger than for the matrix for both cues and for the difference between them. Together these results demonstrate that for the high-probability cue, sessions in which the tone-evoked licking is greater, the neuropil response also is greater, and that this relationship holds for both compartments, but more so for striosomes. For the low-probability cue, there is no relationship between tone-evoked licking and the tone-evoked neuropil responses. Finally, the differential licking response between the two cues predicts the difference in the striosomal neuropil response but not the matrix neuropil response.